# A split and inducible adenine base editor for precise in vivo base editing

Hongzhi Zeng [1], Qichen Yuan [1,7], Fei Peng [2,7], Dacheng Ma[1,7], Ananya Lingineni[3], Kelly Chee[4], Peretz Gilberd [4], Emmanuel C. Osikpa [1], Zheng Sun [2,5] & Xue Gao [1,3,6]

DNA base editors use deaminases fused to a programmable DNA-binding protein for targeted nucleotide conversion. However, the most widely used TadA deaminases lack post-translational control in living cells. Here, we present a split adenine base editor (sABE) that utilizes chemically induced dimerization (CID) to control the catalytic activity of the deoxyadenosine deaminase TadA-8e. sABE shows high on-target editing activity comparable to the original ABE with TadA-8e (ABE8e) upon rapamycin induction while maintaining low background activity without induction. Importantly, sABE exhibits a narrower activity window on DNA and higher precision than ABE8e, with an improved single-to-double ratio of adenine editing and reduced genomic and transcriptomic off-target effects. sABE can achieve gene knock-out through multiplex splice donor disruption in human cells. Furthermore, when delivered via dual adeno-associated virus vectors, sABE can efficiently convert a single A•T base pair to a G•C base pair on the *PCSK9* gene in mouse liver, demonstrating in vivo CID-controlled DNA base editing. Thus, sABE enables precise control of base editing, which will have broad implications for basic research and in vivo therapeutic applications.

As an emerging class of precision genome-editing tools, DNA base editors consist of a deaminase fused to a programmable DNA-binding protein, enabling targeted nucleotide conversions without introducing double-stranded DNA breaks[1,2]. Adenine base editors (ABEs) utilize an evolved *Escherichia coli* tRNA adenosine deaminase (TadA) and act on a single-stranded DNA substrate for A•T to G•C base conversions[2], which have been tested in animal models[3–9] and primary human cells[4,8–11] with various applications, including site-directed mutagenesis[12], gene silencing[13], gene knockout[10], gene isoform discovery[14], functional screens of epigenetic markers[15] or pathogenic mutations[16], and molecular recording[17]. ABEs are particularly useful for investigating or therapeutically correcting human pathogenic alleles because nearly half of the disease-causing point mutations could be

corrected by reversing the pathogenic A•T base pair to a G•C base pair[18,19]. Recently, TadA-ABEs have been re-engineered to achieve other types of base editing, including C-to-T[20–22], C-to-G[21], C/A-to-T/G[20,22], or A-to-Y[23] conversions.

However, the lack of precise control over the deamination activity of ABE limits its application in research and therapy. The current TadAs in ABEs are constitutively active, and the uncontrolled deaminase can cause undesirable genomic and transcriptomic off-target effects[24–27], raising concerns for ABEs' application for the production of genetically modified organisms and gene therapy. For instance, BEs lead to both genomic and transcriptomic off-target due to long-term expression in vivo in transgenic mice, and mice zygotes injected with ABE7.10 encoded AAV exhibit low birth rates[28]. Although inducible promoters

[1]Department of Chemical and Biomolecular Engineering, Rice University, Houston, TX 77005, USA. [2]Department of Medicine, Division of Diabetes, Endocrinology and Metabolism, Baylor College of Medicine, Houston, TX 77030, USA. [3]Department of Bioengineering, Rice University, Houston, TX 77005, USA. [4]Department of Biosciences, Rice University, Houston, TX 77005, USA. [5]Department of Molecular and Cellular Biology, Baylor College of Medicine, Houston, TX 77030, USA. [6]Department of Chemistry, Rice University, Houston, TX 77005, USA. [7]These authors contributed equally: Qichen Yuan, Fei Peng, Dacheng Ma. ✉e-mail: zheng.sun@bcm.edu; xue.gao@rice.edu

can be used to regulate the expression of ABEs[29,30], the leaky expressions and the delayed response from transcription to translation are highly undesirable. Post-translational inducible control of Cas proteins[31–34] can potentially regulate ABE recruitment to the genome but still cannot directly control the deaminase activity of ABE, which does not curtail its off-target effects[25–27,35]. Thus, precision control of the deaminase activity of ABEs would greatly expand its applications.

Here, we present a split ABE (sABE) design with inducible deaminase activity by integrating the chemically induced dimerization (CID) system[36]. We demonstrate that ABE8e[24] can be split into two inactive parts in the TadA-8e deaminase domain: one fused to FK506-binding protein 3 (FKBP3) and the other to FKBP-rapamycin binding (FRB) protein[37]. These two ABE components can reassemble into an active form upon rapamycin-induced FRB-FKBP3 heterodimerization. Through extensive engineering and optimization, we engineer sABE v3.22, which shows efficient and precise on-target single adenine editing upon rapamycin induction and significantly reduced genomic and transcriptomic off-target effects. Using dual adeno-associated viral (AAV) vectors to deliver sABE v3.22 in mice, we perform inducible editing of the *PCSK9* gene and demonstrate high precision on the targeted adenine, showcasing in vivo CID-controlled DNA base editing.

## Results

### Chemically inducible split ABE (sABE) with tightly regulated deaminase activity

To monitor the DNA deaminase activity, we created a fluorescence reporter by introducing a premature stop codon into the EYFP gene via a C•G to T•A base pair conversion, rendering it dysfunctional (EYFP*) (Fig. 1a). ABE guided by a single guide RNA (sgRNA) can edit the adenine on the antisense strand of the EYFP* gene and convert the A•T base pair back to the G•C base pair, thereby restoring the original glutamine codon and resulting in the expression of full-length, functional EYFP (Fig. 1a). We validated the response of EYFP fluorescence to ABE8e in HEK293T cells, without detectable background fluorescence in the absence of ABE8e (Fig. 1a).

To achieve inducible control over ABE deaminase, we split TadA-8e into two inactive parts (TadA-8e$_N$ and TadA-8e$_C$) and fused each part to FRB and FKBP3, respectively. In the presence of rapamycin, FKBP3 and FRB will heterodimerize, bringing the two parts of TadA-8e into proximity and enabling their assembly into a functional unit (Fig. 1b). We constructed sABE v1 and sABE v2 by splitting the TadA-8e[38] deaminase into two fragments. The split sites occurred at loop-25 for sABE v1 and loop-74 for sABE v2 (Fig. 1d). An FKBP3-FRB dimer insertion into these peripheral flexible loop regions is unlikely to alter the TadA-8e core catalytic domain or the reassembly of TadA-8e$_N$ and TadA-8e$_C$. In sABE v1, TadA-8e$_N$ contains the first 24 amino acids of the TadA-8e, which is linked to an FRB via a flexible linker to its C-terminus (Supplementary Fig. 2a). We also fuse a bipartite SV40 nuclear localization signal (NLS) at the N-terminus of TadA-8e$_N$. TadA-8e$_C$ contains the remaining 142 amino acids of the TadA-8e and is fused to an FKBP3 at its N terminus and a *Streptococcus pyogenes* Cas9 nickase (nSpCas9, D10A) at its C terminus. Each terminus has a monopartite SV40 NLS. These two components, sABE(N) and sABE(C), are expressed separately from two plasmids under a cytomegalovirus promoter (pCMV). The constructs in sABE v2 are similar, except that the split site occurs after Arginine 74 of TadA-8e. We co-transfected HEK293T cells with plasmids encoding sABE(N), sABE(C), EYFP*, and sgRNA, using EBFP as a negative control, followed by induction of the sABE activity with 100 nM rapamycin 12 h after transfection (Fig. 1c). Forty-eight hours after induction, we quantified the normalized fluorescence intensity and the percentage of EYFP-positive cells by flow cytometry (Fig. 1c, Supplementary Fig. 1). sABE v1 and v2 successfully activated the EYFP reporter upon rapamycin induction, with sABE v2 showing higher EYFP activation but also higher background (Fig. 1h, Supplementary Fig. 2b). To further investigate split sites adjacent to Arginine 74, we created

sABEs v2.1 to v2.4 by shifting the split site one amino acid at a time (Supplementary Fig. 2c). We found that sABE v2.3, with the split occurring after Isoleucine 76 of TadA-8e, had higher EYFP activation upon rapamycin induction and lower background compared to sABE v2 (Supplementary Fig. 2d).

To further improve the rapamycin-induced deaminase activity and reduce the background activity under the non-induced condition, we optimize components in the sABE construct (Fig. 1e). First, we developed sABE v2.7 by adding a nucleoplasmin NLS to the C terminus of sABE v2.3(N) while keeping the same sABE v2.3(C). Although this modification enhanced editing efficiency, we also observed a higher background activity, probably due to the auto-reassembly of the TadA-8e fragments in the nucleus when they are abundant (Fig. 1h, Supplementary Fig. 3b). Next, we characterized the effect of the dimerization domain copy number on sABE activity. We constructed sABE v2.8, v2.9, and v3.11 by introducing an additional copy of the dimerization domains based on sABE v2.3 (Supplementary Fig. 2e). We found that sABE v3.11, harboring two copies of FRB domain at the C terminus of sABE(N) and two copies of FKBP3 domain at the N terminus of sABE(C), led to a comparable level of EYFP reporter activation with a reduced background (Supplementary Fig. 2f). We then tested different types of linkers with varying lengths between TadA-8e$_N$ and 2×FRB domain in sABEv3.11(N) and between 2×FKBP3 domain and TadA-8e$_C$ in sABE v3.11(C), creating four versions of sABE(N) and four versions of sABE(C). We transfected different combinations of resulting sABE(N) and sABE(C) constructs and screened a total of 16 sABEs (v3.11 to v3.44) using our fluorescence reporter assay (Supplementary Fig. 3a). We chose sABE v3.22 as the final version since it showed comparable EYFP activation with sABE v3.11 while exhibiting significantly reduced background activity under the non-induced condition (Fig. 1f).

Further, after evaluating a range of rapamycin concentrations, we found that 100 nM rapamycin effectively activated sABE v3.22 (Fig. 1g, Supplementary Fig. 3d). We decided to use this concentration for subsequent experiments. In addition, we selected five sABEs (v1, v2, v2.3, v2.7, and v3.11) to compare reporter assay responses and endogenous gene editing efficiencies at three genomic sites in HEK293T cells. The results showed a strong correlation between the sABE activities in these two assays (Fig. 1h, i, Supplementary Fig. 3b, c). We also examined whether the sABE system could be deactivated. Cells transfected with sABE v3.22 were treated with 10, 25, 50, or 100 nM rapamycin for 2 h, after which the culture medium was changed to remove the rapamycin. Both the reporter assay and genomic editing data showed sABE v3.22 activation in rapamycin-treated groups. The group from which rapamycin was removed showed decreased deaminase activity compared to the rapamycin-sustained group (Supplementary Fig. 4). This effect was less significant when the initial concentration of rapamycin was increased beyond 50 nM, likely due to the residual intracellular rapamycin and the inefficient excretion and degradation of rapamycin in HEK293T cells in vitro[39]. In sum, we successfully split the ABE8e into two inactive parts at the TadA-8e deaminase domain and rendered its deaminase activity chemically inducible using the FKBP3-FRB CID. Through engineering approaches and fluorescence reporter screening, we developed sABE v3.22, which has a high level of induced base editing activity and a low level of non-induced background activity.

### sABE v3.22 achieves high DNA on-target editing efficiencies and enhanced precision

We compared the performance of sABE v3.22 to the intact ABE8e by targeting 19 human genomic loci that span different sequence contexts (Fig. 2a, Supplementary Fig. 5a). ABE8e achieved A-to-G conversions ranging from 7.2% to 72% in the conventional A$_4$-A$_8$ activity window, with a mean of 56% at the A$_4$-A$_5$ positions. In the absence of rapamycin, sABE v3.22 showed very low background A-to-G

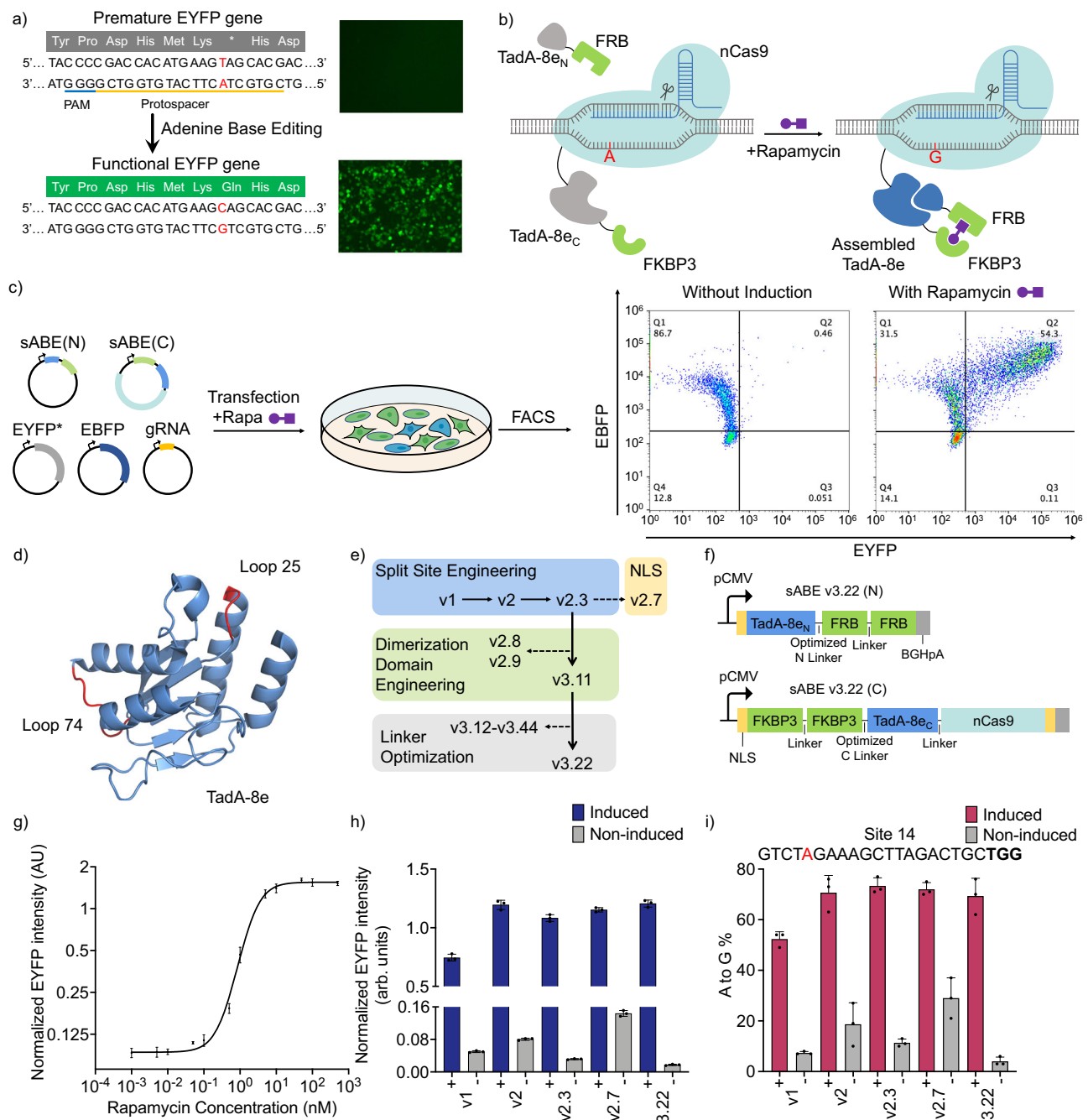

**Fig. 1 | Chemically inducible split ABE (sABE) with tightly regulated deaminase activity. a** Schematic of the EYFP fluorescence. An A-to-G conversion on the highlighted adenine on the antisense strand of the dysfunctional EYFP* gene can restore the expression of functional EYFP protein. **b** Schematic of the sABE. In the absence of rapamycin (Left), two parts of the TadA-8e: TadA-8e$_N$ and TadA-8e$_c$, remain inactive. In the presence of rapamycin (Right), the rapamycin-FKBP3 complex binds to FRB, bringing the TadA-8e$_N$ and TadA-8e$_c$ to spatial proximity to form an active ABE unit. nCas9: *Streptococcus pyogenes* Cas9 (D10A) nickase; FKBP: FK506 binding proteins; FRB: FKBP-rapamycin binding domains. **c** EYFP* reporter assay in HEK293T cells. Cells are co-transfected with five plasmids, using EBFP as a transfection control. Representative FACS data show the EYFP* activation by sABE in the presence of rapamycin. **d** Crystal structure of the TadA-8e deaminase domain of ABE8e (PDB: 6VPC[38]). Highlighted loop-25 and loop-74 regions indicate where the TadA-8e is split into two parts for the sABE v1 and v2, respectively. **e** Engineering steps to increase the rapamycin-induced deaminase activity and to decrease the non-induced background. **f** Diagram of the sABE v3.22 constructs. pCMV: cytomegalovirus promoter. **g** Dosage-response curve of the reporter assay to sABE v3.22. Normalized EYFP intensity is the mean EYFP intensity divided by the mean EBFP intensity. **h** EYFP* reporter responses to five versions of sABEs. Blue: with 100 nM rapamycin induction; gray: non-induced. **i** A-to-G editing efficiencies of the highlighted adenine by five versions of sABEs at Site 14. Red: with 100 nM rapamycin induction; gray: non-induced. Editing efficiencies in (**i**) are evaluated by Sanger sequencing. Dots represent data from three independent biological replicates, and bars represent their mean with s.d.

conversions in the A$_4$-A$_8$ window ranging from 0.1% to 3.1%, with a mean of 0.7%. The deaminase activity of sABE v3.22 was induced by an average of 89-fold (ranging from 15-fold to 389-fold), reaching a mean of 80% (ranging from 53% to 97%) of the activities of intact ABE8e at A$_4$-

A$_5$ positions (Fig. 2b). Additionally, sABE v3.22 exhibited a narrower activity window of A$_4$ and A$_5$, with reduced activity on A$_6$ and A$_7$ and minimum activity elsewhere in the protospacer (Fig. 2c, Supplementary Fig. 5b).

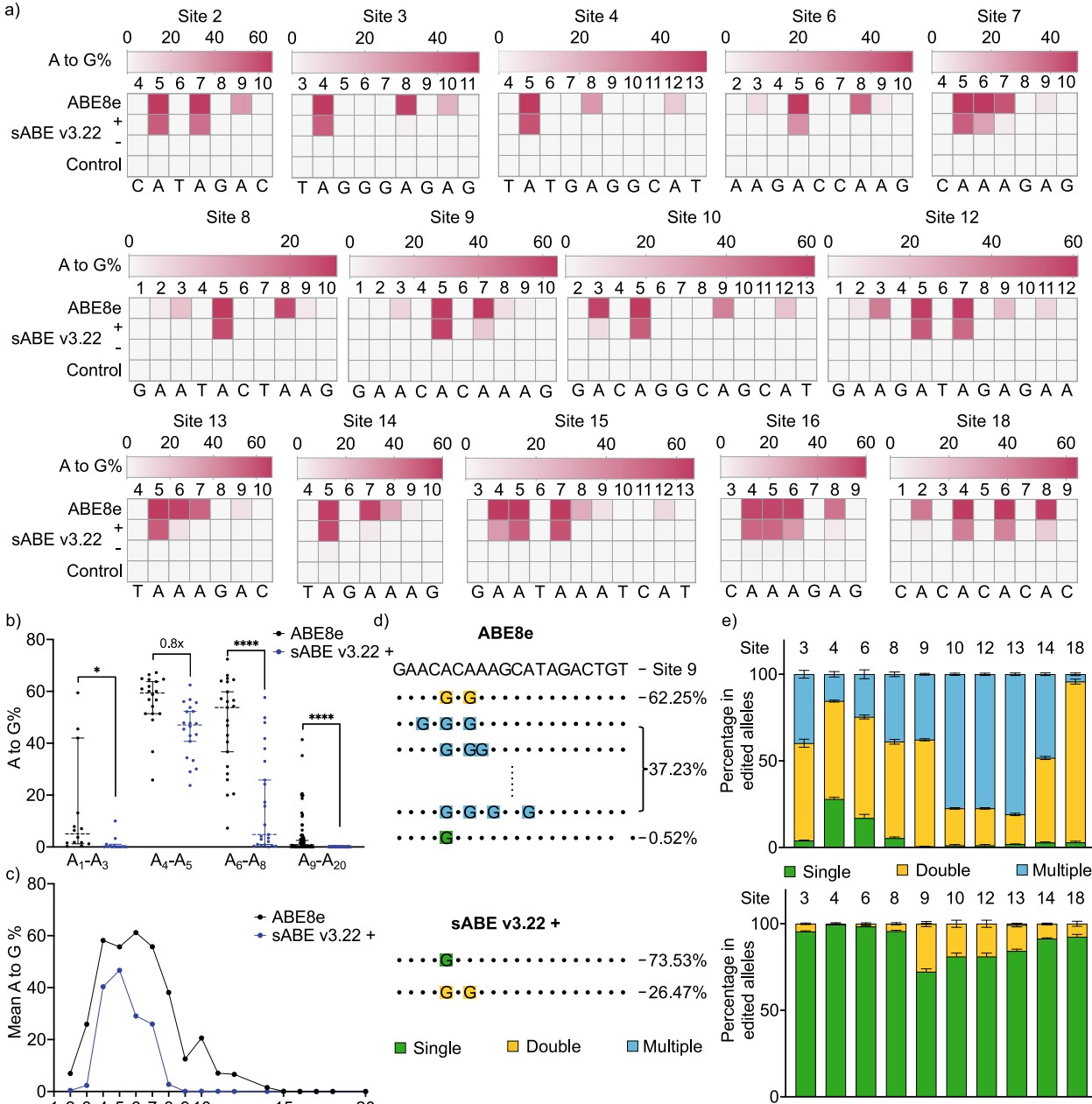

**Fig. 2 | sABE v3.22 achieves high DNA on-target editing efficiencies and enhanced resolutions. a** Heat maps show the on-target DNA A-to-G editing efficiencies of ABE8e, sABE v3.22 induced with 100 nM rapamycin, and non-induced sABE v3.22 within the conventional $A_4$-$A_8$ activity window (numbered with 1 as the most PAM-distal position) at fourteen genomic loci. The control group was mock-transfected. Color brightness represents the mean value of three independent biological replicates. **b** Mean A-to-G editing efficiencies of ABE8e and sABE v3.22 at nineteen tested loci. Each dot represents the mean value of A-to-G editing efficiencies on adenine at a tested locus in **a** and Supplementary Fig. 5a. Dots are divided into bins based on the position of their corresponding adenine in the protospacer. $A_1$-$A_3$: $n = 14$, $P = 0.0258$; $A_4$-$A_5$: $n = 20$; $A_6$-$A_8$: $n = 23$, $P < 0.0001$; $A_9$-$A_{20}$: $n = 62$, $P < 0.0001$. **c** Comparison of the activity window and base editing efficiencies between ABE8e and sABE v3.22. **d** Representative sequence reads containing A-to-G conversions in ABE8e edited alleles (Top) or in sABE v3.22 edited alleles (Bottom) at Site 9. Colored nucleotides represent A-to-G conversions, and their colors represent if the reads contain single (green), double (yellow), or multiple (blue) A-to-G conversions. Numbers on the side represent the percentage of corresponding reads in total reads containing A-to-G conversions. **e** Bar plot showing the percentage of reads containing different numbers of base conversions in ABE8e edited alleles (Top) or in sABE v3.22 edited alleles (Bottom) at ten genomic loci. Bars represent the mean ± s.d of three individual biological replicates. **b** uses the unpaired two-tailed *t*-test; ns, not significant; *$P < 0.05$; **$P < 0.01$; ***$P < 0.001$; ****$P < 0.0001$.

As a result of the narrower activity window of sABE v3.22, we observed a significant change in the distribution of reads with A-to-G conversions. For example, sABE v3.22 and ABE8e achieved comparable (62% and 65%) A-to-G conversion on $A_5$ at Site 9. However, of all reads with A-to-G conversions from the intact ABE8e-transfected samples, only 0.53% had single $A_5$ editing. Over 99% had $A_5$ and $A_7$ double edits or more than two A-to-G conversions across the seven adenines

between positions 2 and 12 (Fig. 2d). In contrast, in the sABE v3.22-transfected samples, 74% of all reads with A-to-G conversions had single $A_5$ editing, 26% showed $A_5$ and $A_7$ editing, and there was no editing at more than two adenines. At nine out of the 19 sites tested, the ratio of single adenine edits increased from <25% to >73% when using sABE v3.22 (Fig. 2e). At the other ten sites, the ratios of single and double edits increased, while the ratio of multiple edits decreased

significantly (Supplementary Fig. 5c). Taken together, the sABE v3.22 system demonstrates higher precision and reduced bystander editing compared to ABE8e, which would allow for more precise single adenine editing.

We subsequently constructed and compared the performance of sABEs with different TadA variants, including sABE(V106W) v3.22 and sABE(F148A) v3.22, as V106W[25] and F148A[27] are beneficial mutations that decrease TadA transcriptomic off-target effects. Among the eight genomic sites tested, the sABE v3.22 demonstrated a mean A-to-G conversion rate of 62% (ranging from 30% to 82%), achieving 94% (ranging from 57% to 113%) of the intact ABE8e activity (ranging from 52% to 76%) with an average induction of 18-fold (ranging from 4-fold to 48-fold) (Supplementary Fig. 6a). sABE(V106W) v3.22 showed an average A-to-G conversion rate of 50% (ranging from 22% to 69%), achieving 73% (ranging from 38% to 97%) of the intact ABE8e(V106W) activity (ranging from 56% to 76%) with an average induction of 28-fold (ranging from 4-fold to 83-fold) (Supplementary Fig. 6b). sABE(F148A) v3.22 exhibited an average of 57% A-to-G conversion rate (ranging from 22% to 69%) among these sites, achieving 82% (ranging from 56% to 97%) of the ABE8e(F148A) activity (ranging from 60% to 77%), with an average induction fold of 24-fold (ranging from 4-fold to 77-fold) (Supplementary Fig. 6c). Consistent with sABE v3.22, both V106W and F148A variants demonstrated a narrower editing window, with peak activity at the $A_4$ and $A_5$ positions.

We further explore the compatibility of sABE v3.22 by replacing the nSpCas9 with the more compact *Staphylococcus aureus* Cas9 nickase (nSaCas9)[24,40]. At the two genomic sites tested, sSaABE8e demonstrated editing efficiencies of 44% and 13%, respectively. We also exhibited an average of 30-fold induced activity compared to the non-induced group (Supplementary Fig. 7a). Under optimized assay conditions, TadA8e, when coupled with the engineered dead Cas12f variants from an uncultured archaeon (Un1Cas12f1), specifically Cas-MINI v3.1 and CasMINI v4[41–43], along with engineered sgRNA scaffold ge4.1[44], resulted in 2-4% A-to-G conversion rates across the three sites examined (Supplementary Fig. 7b, c). The corresponding split systems, sCasMINI v3.1 and sCasMINI v4 showed 1-2% A-to-G conversion rates with undetectable background (Supplementary Fig. 7b, c). Together, these data demonstrate that the sABE v3.22 architecture paired with SpCas9 showed superior editing efficiency and is compatible with other TadA variants bearing beneficial mutations, such as V106W and F148A. sABE v3.22 architecture is also compatible with smaller Cas domains, including SaCas9 and engineered Un1Cas12f1.

## Genomic and transcriptomic off-target effects in mammalian cells

We analyzed genomic off-target effects of sABE v3.22 and ABE8e in HEK293T cells at Cas9-dependent DNA off-targets that have been reported[11] or predicted using Cass-OFFinder[45] (Fig. 3b). We detected A-to-G conversions at 13 out of the 15 analyzed off-target sites. In the absence of rapamycin, sABE v3.22 exhibited low non-induced A-to-G conversions at on-target sites (mean 1.2%) and at off-target sites (mean 0.52%) within the $A_4$-$A_8$ window (Supplementary Fig. 8a). With 100 nM rapamycin induction, sABE v3.22 showed a mean of 8.6% off-target A-to-G conversions within the $A_4$-$A_8$ activity window, decreasing the Cas9-dependent off-target effects by >75% compared to the intact ABE8e (mean 35%), and resulting in 1.8 - 130-fold increases in the on-to-off-target ratio (Fig. 3a, c). Furthermore, due to the narrower activity window of sABE v3.22, we observed no A-to-G conversion outside the $A_4$-$A_8$ window on Cas9-dependent off-targets (Supplementary Fig. 8a).

Next, we characterized Cas9-independent off-target effects of sABE v3.22 and ABE8e using the previously established orthogonal dSaCas9 R-loop assay[46] (Fig. 3d). In this assay, HEK293T cells are cotransfected with plasmids containing sABE v3.22 constructs and a SpCas9 sgRNA specific for the desired on-target site, along with additional plasmids encoding a dead SaCas9 (dSaCas9) and a

SaCas9 sgRNA which is orthogonal to the SpCas9 sgRNA and targets an unrelated genomic locus[46].In this setup, dSaCas9 unwinds the DNA double helix to reveal single-stranded DNA, which can serve as a substrate for the deaminase fused to nSpCas9, independently of SpCas9 binding. Thus, Cas9-independent off-target effects could be determined by measuring A-to-G conversion rates at this genomic locus unrelated to the SpCas9 target sequences. We observed Cas9-independent off-target A-to-G conversions by intact ABE8e ranging from 0.87% to 9.2% across the five tested orthogonal R-loops (Fig. 3f). sABE v3.22 reduced these off-target activities to undetectable levels in three orthogonal R-loops and <0.36% in the other two sites. Meanwhile, DNA on-target editing efficiencies remain comparable between sABE v3.22 and ABE8e (Fig. 3e, Supplementary Fig. 8b). The non-induced group showed no difference from the mock-transfected control (Fig. 3e, Supplementary Fig. 8b). Additionally, we repurposed our EYFP* reporter assay to detect Cas9-independent off-target effects by cotransfecting ABEs with dSaCas9 and a SaCas9 sgRNA that aims to form an R-loop at the premature stop codon site. Consistent with the genomic R-loop assay, we detected activated EYFP fluorescence in the intact ABE8e-transfected HEK293T cells but not in sABE v3.22-transfected cells (Supplementary Fig. 8c, d), indicating lower Cas9-independent off-target editing using sABE v3.22.

To compare the extent of transcriptomic off-target effects of sABE v3.22 and ABE8e (Fig. 3g), we transfected HEK293T cells with plasmids encoding ABE8e-P2A-EGFP, sABE v3.22(C)-P2A-EGFP-P2A-sABE v3.22(N), or nCas9(D10A)-P2A-EGFP (Supplementary Fig. 9a). Each plasmid also encodes a sgRNA targeting Site 11. We sorted the transfected cells with the top 5% mean fluorescence intensities and extracted their RNA and DNA for high-throughput sequencing or Sanger sequencing (Supplementary Fig. 10). In sorted cells, rapamycin-induced sABE v3.22 achieved comparable on-target DNA editing (mean $A_5$ 73%) to ABE8e (mean $A_5$ 79%). Non-induced sABE v3.22 showed a higher background activity (mean $A_5$ 20%) when compared to those observed in previous experiments since the sorted cells had the highest expression of ABEs, indicated by their EGFP fluorescence intensities (Fig. 3h, Supplementary Fig. 9b). Using the Genome Analysis Toolkit[47] (GATK) best practices for variant calling and further downstream filtering, we identified mRNA nucleotide positions that were altered in cells expressing ABE8e, sABE v3.22, or nCas9 but not in the mock-transfected controls (details in Methods). We found a significant increase in transcriptome-wide A-to-I single nucleotide variations in ABE8e-transfected HEK293T cells (mean 24,670) compared to the nCas9(D10A)-transfected cells (mean 125) (Fig. 3i, Supplementary Fig. 9c). Meanwhile, sABE v3.22 reduced 70% of transcriptome A-to-I mutations, with a mean of 7,279 A-to-I conversions called. Without rapamycin induction, the number of transcriptional A-to-I mutations in the sABE v3.22-transfected cells (mean 149) was similar to that in nCas9(D10A)-transfected cells (mean 125). In summary, these data suggest that the small-molecule-controlled sABE v3.22 maintains a comparable level of on-target activity with reduced genomic and transcriptomic off-target effects.

## Inducible multiplex gene knockouts in mammalian cells

ABEs can achieve gene knockout by targeting gene splice donor regions, leading to disrupted pre-mRNA splicing processes, such as exon skipping, intron inclusion, and cryptic splice-site utilization[10,48]. To assess the performance of sABEv3.22 for inducible gene knockout, we targeted two genes expressing Beta-2 microglobulin (B2M) or CD46 regulatory proteins that have been widely studied in the context of allogeneic cell therapies and cancer research[49–52]. We utilized our recently reported drive-and-process multiplex base editing (DAP-MBE) array[53] to express multiple sgRNAs that guide ABEs to disrupt B2M or CD46 splice donors (Fig. 4a). To compare sABE v3.22 and intact ABE8e, we co-transfected HEK293T cells with the DAP-MBE array expressing two sgRNAs targeting B2M splice donors and either sABE v3.22 or

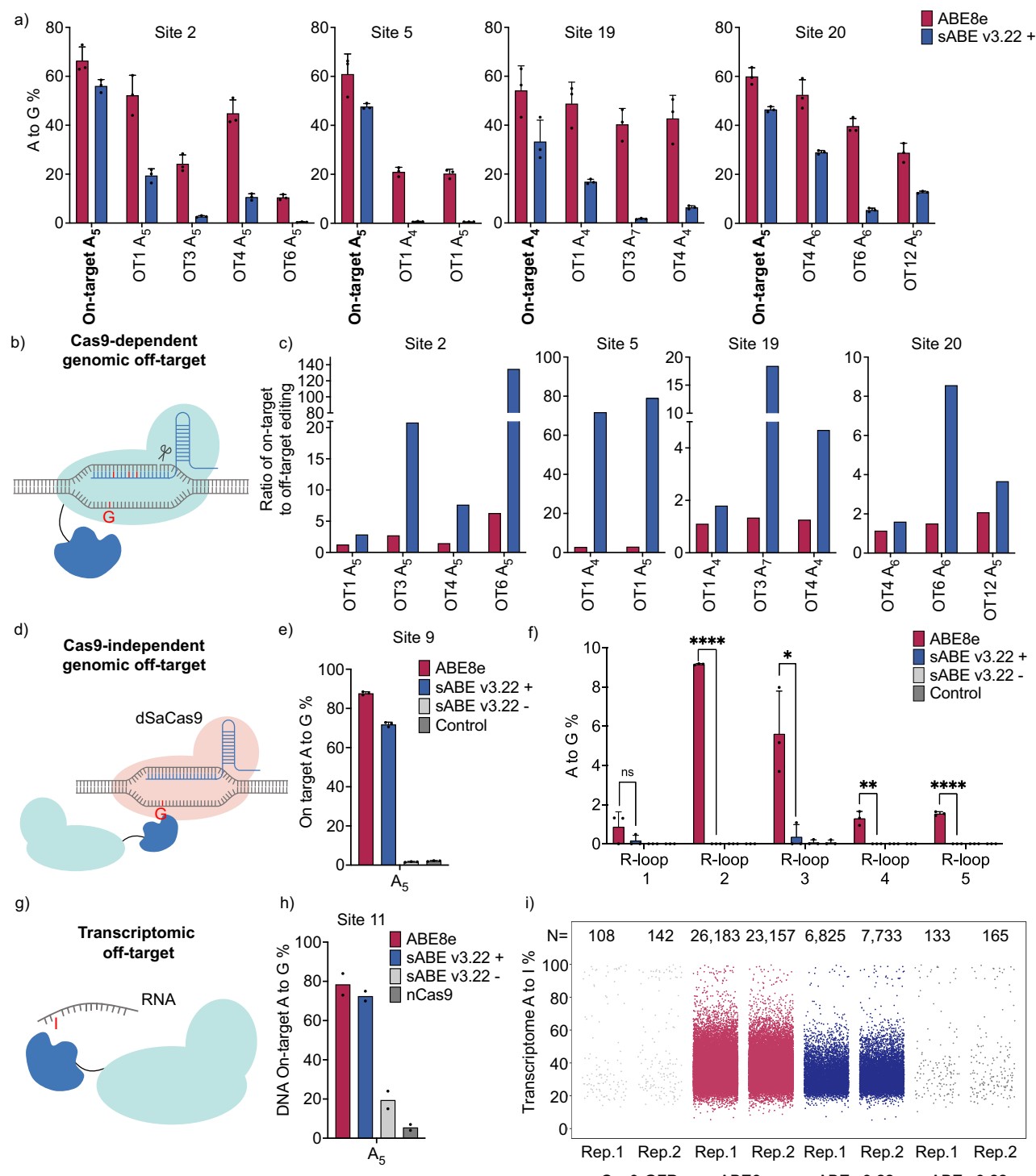

**Fig. 3 | Genomic and transcriptomic off-target effects in mammalian cells. a** A-to-G editing efficiencies on adenine in the conventional $A_4$-$A_8$ editing window in HEK293T cells by ABE8e or sABE v3.22 at four DNA on-target loci and eleven Cas9-dependent DNA off-target loci. OT: off-target. $n$ = 3 **b** Schematic of Cas9-dependent genomic off-target effects by ABEs. **c** The ratio of on-target to off-target A-to-G editing efficiencies using data from **a**. **d** Schematic of orthogonal R-loop assay for detecting Cas9-independent DNA genomic off-target effects by ABEs. dSaCas9: dead *Staphylococcus aureus* Cas9. **e** DNA on-target A-to-G editing efficiencies in the R-loop assay by ABE8e or sABE v3.22. $n$ = 3. **f** Cas9-independent off-target DNA A-to-G conversions detected by the orthogonal R-loop assay. High-throughput sequencing reads consisting of <0.2% of total reads were not considered. $n$ = 3. $P$-values from left to right are 0.199873, <0.000001, 0.015994, 0.003168, and 0.000012. **g** Schematic of transcriptomic off-target effects by ABEs. **h** DNA on-target A-to-G editing efficiencies in the RNA-seq experiment, evaluated by Sanger sequencing. $n$ = 2 **i** Jittered strip plots representing frequencies of A-to-I and T-to-C variations identified from RNA-seq experiments where HEK293T cells were transfected with ABE8e, sABE v3.22, or nCas9. Each dot represents an individual A-to-I or T-to-C variation at an individual nucleotide. T-to-C variations were considered as A-to-I variations on the Crick strand because strand-sensitive RNA-seq data were aligned to only the Watson strand. N is the total number of A-to-I and T-to-C single nucleotide variations identified. Rep: individual biological replicate. In **a**, **e**, **f**, and **h**, dots represent individual biological replicates, and bars represent mean ± s.d. In **c**, bars represent the ratio of mean values of on-target efficiencies to mean values of off-target efficiencies using data from **a**. **f** Uses the multiple unpaired two-tailed $t$-test; ns, not significant; *$P$ < 0.05; **$P$ < 0.01; ***$P$ < 0.001; ****$P$ < 0.0001.

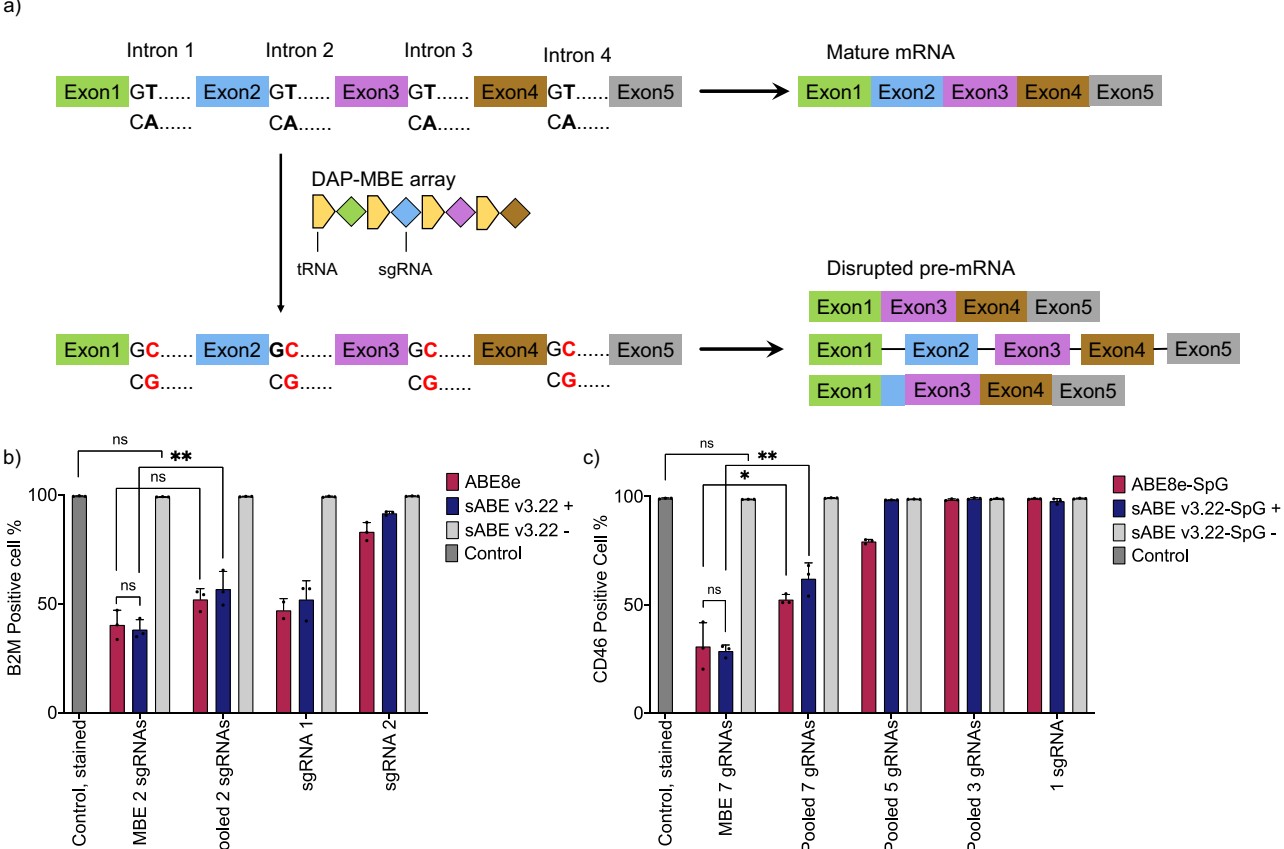

**Fig. 4 | Inducible multiplex gene knockouts in mammalian cells. a** Schematic of the gene knockout by introducing A•T to G•C base conversions (highlighted in red) at multiple splice donor sites using DAP-MBE and ABEs. The consensus sequence of splice donors is GT. The target adenines are located on the antisense strand and are emboldened. DAP-MBE: drive-and-process multiplex base editing array. **b, c** Protein expression of **b** B2M gene and **c** CD46 gene in HEK293T cells transfected with sgRNAs or DAP-MBE array targeting splice donors and sABE v3.22 or ABE8e. The percentages of B2M- or CD46-positive cells were quantified using antibody-based FACS analysis. $n = 2$ in **b** in the group transfected with ABE8e and sgRNA 1. $n = 3$ in all other groups. Dots represent individual biological replicates, and bars represent mean ± s.d. **b, c** use the unpaired two-tailed *t*-test; ns, not significant; *$P < 0.05$; **$P < 0.01$; ***$P < 0.001$; ****$P < 0.0001$. *P*-values in **b** from left to right are 0.0502 (ns), 0.6709 (ns), 0.0709 (ns), and 0.0252 (**). *P*-values in **c** from left to right are 0.1024 (ns), 0.7595 (ns), 0.0286 (*), and 0.0018 (**).

ABE8e plasmids, followed by 100 nM rapamycin treatment 12 h later. Cells were cultured for another 8 days for B2M protein degradation and cell division before antibody-based FACS analysis (Supplementary Fig. 11, 12). With rapamycin induction, sABE v3.22 achieved over 60% knockout efficiency for B2M, which was similar to intact ABE8e (Fig. 4b)[53]. There was no difference between the mock-transfected group and the non-induced group transfected with sABE v3.22 and DAP-MBE array (Fig. 4b). We also compared the DAP-MBE array with individual sgRNAs or pooled sgRNAs delivered from two plasmids. Targeting both splice donors resulted in a higher B2M knockout rate compared to targeting only the B2M intron 1 splice donor (mean 48% by sABE v3.22 and 53% by ABE8e) or only the B2M intron 2 splice donor (mean 9% by sABE v3.22 and 17% by ABE8e) (Fig. 4b, Supplementary Fig. 13a). The DAP-MBE strategy led to more efficient B2M knockout compared to pooled sgRNAs strategy.

Among the four CD46 splice donors with nearby NGG sequences, only two can be targeted with the target adenine in the ABE activity window. Therefore, we constructed ABE8e-SpG and sABE v3.22-SpG by integrating the recently reported SpCas9 variant SpG[54] with a relaxed NGN PAM requirement instead of NGG. Through this approach, we were able to target five additional splice donors. We found that multiplex disruption of seven CD46 splice donors using the DAP-MBE array led to the highest CD46 knockout efficiency, with a mean of 71% by sABE v3.22 and 69% by ABE8e-SpG (Fig. 4c). The group with non-induced sABE v3.22-SpG showed no statistical difference from the mock-transfected group. Similar to the B2M knockout results, disrupting fewer

CD46 splice donors was less efficient than disrupting all seven targetable splice donors (Fig. 4c, Supplementary Fig. 13b). The DAP-MBE strategy also led to more efficient CD46 knockout compared to the pooled sgRNAs strategy (38% by sABE v3.22-SpG and 48% by ABE8e).

## Inducible in vivo editing of mouse *PCSK9* gene
To explore the in vivo application of sABE v3.22, we packaged it into three AAV vectors. We split the sABE(C) into two parts before lysine 468 in the nCas9 domain, fused each part to the corresponding split moiety of gp41-1 intervening protein (intein)[55], and packaged them into separate AAV vectors. The sABE(N) and a sgRNA targeting Site 9 were packaged into a third AAV vector (Fig. 5a). When delivered into HEK293T cells via triple AAV vectors, the sABE v3.22 achieved 40% on-target $A_5$ editing with 100 nM rapamycin induction and showed 4.2% $A_5$ background activity without induction (Fig. 5b, Supplementary Fig. 14a). Notably, of the sequencing reads with A-to-G conversions, 92% showed single $A_5$ editing (Supplementary Fig. 14b). Similarly, when the sABE v3.22 system was delivered into HEK293T cells via three lentiviral vectors (Supplementary Fig. 14c), we observed 43% on-target $A_5$ editing with 100 nM rapamycin induction and 3.0% $A_5$ background activity without induction (Supplementary Fig. 14d). Consistently, 86% of sequencing reads with A-to-G conversions showed single $A_5$ editing (Supplementary Fig. 14e). These results demonstrate the compatibility of our sABE system with both viral platforms for gene delivery.

The proprotein convertase subtilisin-like kexin type 9 (*PCSK9*) gene is an attractive target for treating atherosclerotic cardiovascular

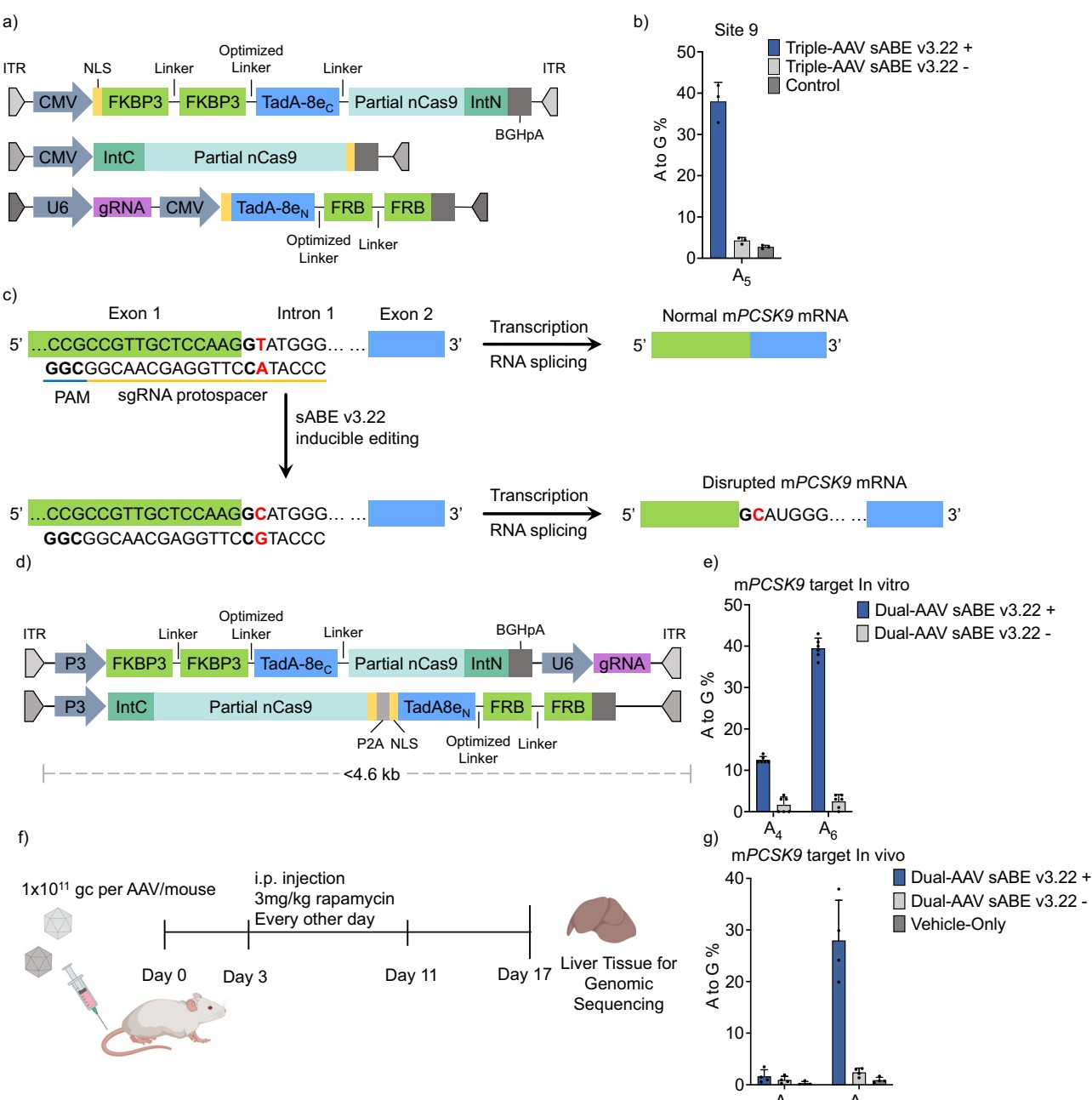

**Fig. 5 | Inducible in vivo editing of mouse *PCSK9* gene. a** Triple AAVs encoding sABE v3.22. IntN and IntC refer to the N- and C-terminal parts of the gp41-1 intein, respectively. **b** A-to-G base editing efficiencies at Site 9 A₅ in HEK293T cells transduced with triple AAVs encoding sABE v3.22. $n = 3$. **c** Schematic of ABE-mediated m*PCSK9* gene knockout. The intron 1 splice donor is emboldened, and the target adenine is highlighted in red. **d** Dual AAVs encoding sABE v3.22 for in vivo editing. **e** A-to-G editing efficiencies at m*PCSK9* intron 1 splice donor site in the HEK293T stable cell line integrated with a 200-bp m*PCSK9* exon 1-intron 1 junction. sABE v3.22 was induced with 200 nM rapamycin. Editing efficiencies were assessed

14 days after the dual-AAV transduction. $n = 6$. **f** Schematic outline of the mouse experiment. Vectors in **d** were administered intravenously via tail vein injection into C57BL/6 J mice. 3 mg/kg rapamycin was dosed via i.p. injection every other day four times. **g** A-to-G base editing efficiencies at m*PCSK9* intron 1 splice donor site in mouse liver tissue. $n = 4$ animals in each group. Vehicle-Only: Mock-injected with the vehicle. High-throughput sequencing reads consisting of <0.2% of total reads were not considered. Dots represent individual biological replicates, and bars represent mean ± s.d.

diseases[4,5,56]. Adenine base editing of the *PCSK9* gene has been shown to lower cholesterol in vivo durably[4,5]. To test inducible gene editing on the *PCSK9* gene in mice, we rearranged and incorporated the sABE v3.22 components into two AAV vectors, targeting the mouse *PCSK9* (m*PCSK9*) intron 1 splice donor (Fig. 5c). The first vector carried the first half of sABE v3.22(C) driven by a pP3 liver-specific promoter[57] and the sgRNA driven by a pU6 promoter. The second vector carried the

other half of sABE v3.22(C), linked to the sABE v3.22(N) through a P2A self-cleaving peptide and promoted by pP3 (Fig. 5d). We packaged the sABE v3.22 into AAV serotype 1 and transduced it into a HEK293T stable cell line integrated with a 200-bp m*PCSK9* gene fragment containing the m*PCSK9* intron 1 splice donor. We observed a mean of 40% targeted A₆ edit with 13% bystander A₄ edit 14 days after transduction, while the non-induced activity was <2.5% (Fig. 5e).

To ensure tissue-specific delivery of the sABE v3.22 system in vivo, we packaged the dual-AAV sABEv3.22 into AAV8, a serotype with high tropism for hepatocytes[58]. We delivered $1 \times 10^{11}$ genome copies (gc) of each AAV to 15-week-old C57BL/6 mice via tail-vein injection (Fig. 5f). Three days later, we treated the mice with 3 mg/kg rapamycin every other day for 8 days via intraperitoneal (i.p.) injection. We euthanized the mice 6 days after the last rapamycin dose, isolated genomic DNA from their liver tissue, and analyzed the on-target efficiency. We observed a mean of 2.4% non-induced background targeted $A_6$ A-to-G conversion. With just four doses of rapamycin, the dual-AAV-delivered sABE v3.22 exhibited up to 40% targeted $A_6$ editing with a mean of 1.6% low bystander $A_4$ editing (Fig. 5g, Supplementary Fig. 15a). We also observed no A-to-G conversion above the background at reported Cas9-dependent off-target sites[5] (Supplementary Fig. 15b, c). Thus, the inducible sABE v3.22 is suitable for in vivo applications.

## Discussion

We present a chemically inducible sABE architecture that utilizes rapamycin-induced dimerization of FKBP3 and FRB to control the activity of TadA-8e deaminase. We engineered an sABE construct (v3.22) that exhibits low background activity and, upon induction, achieves high base-editing activity comparable to ABE8e. The sABE v3.22 system demonstrates higher precision than ABE8e, allowing for more precise single adenine editing with significantly reduced off-target effects. In addition, sABE v3.22 architecture is compatible with TadA variants bearing beneficial mutations, including V106W and F148A, that have the potential to further mitigate transcriptomic off-target effects, as well as other Cas effectors with alternative PAMs, such as SaCas9 and engineered Un1Cas12f1. sABE v3.22 enables highly efficient knockout of human endogenous genes, including B2M and CD46, via multiplex disruptions of splice donors. This multiplex strategy could potentially be applied to other genes. Upon packaging into dual-AAV vectors, the sABE v3.22 achieves efficient inducible base-editing of a therapeutically relevant *PCSK9* gene in the mouse liver. We envision that our control of the sABE system could potentially mitigate the risks associated with the prolonged expression of active ABE8e[59]. Thus, sABE greatly expands the capability of inducible gene editing, with broad implications for basic research and in vivo therapeutic applications. The simplicity of sABE v3.22 applications makes it a valuable tool for inducibly introducing precision A-to-G conversions compared to Cas9 nuclease editing[60] and prime editing technologies[61,62].

Given that the split site (isoleucine 76) can accommodate the insertion of 2×FRB and 2×FKBP, we expect the sABE architecture can be adapted to other post-translational control mechanisms that rely on protein dimerization, including FKBP-FRB with rapamycin analogs (rapalogs)[63], chemically inducible split proteins[64], proteolysis targeting chimeric (PROTAC)-chemically induced dimerization systems[65], and light-induced dimerization (LID) systems[66]. Adapting the sABE v3.22 to work with rapamycin-independent dimerization systems could potentially mitigate side effects of rapamycin, such as immunosuppression[63] and *PCSK9* upregulation[67]. Moreover, we envision that our sABE architecture can be extended to other TadA-containing base editors, such as the recently reported TadCBEs[20–22] (cytosine base editors), TadDEs[20,22] (dual base editors), AYBE[23] (an adenine transversion base editor), Td-CGBE[21] (a C-to-G base editor), TALE-ABEs[68] (a mitochondrial base editor), and inlaid ABEs[69,70] (for simultaneous control of Cas and deaminase activity), which will greatly facilitate the high-precision genome editing applications.

In summary, we demonstrate an inducible split ABE system that utilizes small-molecule induced dimerization to regulate the TadA-8e deaminase for controllable, efficient, high-precision, and in vivo applicable A-to-G base editing. Our work expands the scope of inducible genome editing with the potential for broad research and therapeutic applications.

## Methods

### Ethical Statement

All research conducted complies with relevant regulations. Animal experiment protocols were approved by the Institutional Animal Care and Use Committee (IACUC) of Baylor College of Medicine (BCM).

### Molecular cloning

DNA amplifications were performed by PCR using the 2 × Phanta Max master Mix (Dye Plus, Vazyme, P525). Vectors were linearized by PCR or by restriction digestion. A PCR program with 60 °C annealing temperature and 25 cycles was programmed for 20 μl PCR reaction systems with 100 pmol of each primer and 10-50 ng template to amplify DNA fragments > 2 kb; the cycle number was changed to 35 to amplify DNA fragments <2 kb. Gel electrophoresis of the amplified DNA was conducted in 1.5% DNA agarose gel with 0.5 μg/ml UltraPure Ethidium Bromide (Thermo Fisher Scientific, BP1302-10). The small gel region containing the target DNA fragment was excised, and DNA was extracted using QIAquick Gel Extraction Kit (Qiagen, 28704). Golden Gate assembly was used to assemble the DNA fragments in a 10 μL system containing purified DNA fragments, 1 μl 10 × T4 DNA ligase buffer (New England BioLabs, B0202S), 0.5 μl T4 DNA ligase (200U, New England BioLabs, M0202S), and 0.5 μl BsaI-HFv2 (10U, New England BioLabs, R3733S) or Esp3I (5U, Thermo Fisher Scientific, ER0452). The Golden Gate assembly mixture was cycled between 37 °C and 16 °C for 5 min at each temperature for 15 cycles and incubated at 60 °C for 5 min as the last step. Transformations were performed using *Stbl3* competent cells prepared by the Mix & Go! E. coli Transformation Kit (Zymo, T3001).

DNA oligonucleotides were obtained from Integrated DNA Technologies (IDT). Plasmids containing short insertions, such as sgRNA protospacers, were constructed by ligating annealed and phosphorylated oligonucleotides with other amplified DNA fragments through the Golden Gate assembly. A 20 μl annealing system containing 0.2 nmol of each oligonucleotide and 2 μl 10 × T4 DNA ligase buffer (New England BioLabs) was heated up to 95 °C for 5 min, followed by −1 °C/min ramp down to 37 °C. Next, 1 μl annealed oligonucleotides were added to a 10 μL system containing 0.5 μL T4 Polynucleotide Kinase (5U, New England Biolabs, M0201S) and 1 μL T4 DNA ligase buffer (New England BioLabs) and was incubated at 37 °C for 30 min. Finally, 1 μl annealed and phosphorylated oligonucleotides were used for the Golden Gate assembly. ABE8e (#138489) and lentiGuide-Puro (#52963) were obtained from Addgene and used directly or as PCR templates. FKBP3, FRB, and gp41-1 intein were synthesized by gBlocks (IDT).

Plasmids were isolated using buffers from QIAprep Spin Miniprep Kit (Qiagen, 27104) and were filtered and collected from DNA spin columns (Epoch Life Science, 1920-250). Constructs were verified by Sanger sequencing across all assembly junctions and key regions, including the sequences of the deaminase, FKBP, and FRB. The annotated sequences of each key plasmid developed in this work are available in the shared Benchling links (Supplementary Data. 1).

### Mammalian cell culture

HEK293T cells (American Type Culture Collection, CRL-3216) were cultured in Dulbecco's Modified Eagle's Medium (DMEM) plus Gluta-MAX (Gibco, 10569044) supplemented with 10% (v/v) fetal bovine serum (Gibco, 10437028) and 1% (v/v) penicillin-streptomycin (Gibco, 15140122), hereafter referred as the complete media, in 10 cm TC treated cell culture dish with vents (Greiner Bio-One, 664160). Cells were grown at 37 °C in 5% CO2 incubators and passaged upon reaching 80% confluency.

### Transfection

Cells of low passage number (1–10, passage number of freshly thawed cells is counted as 0) were counted by Countess II FL (Thermo Fisher Scientific) and plated at $2 \times 10^4$ cells per 100 μl complete media per well

for reporter experiments or $0.75 \times 10^4$ cells per 100 µl complete media per well for genomic editing experiments in 96-well plates (Corning, 3598) 16-20 h before transfections. The seeded plate was incubated at room temperature for 15 min before being placed into the incubator. For each well on the plate, transfection plasmids and 0.5 µL Lipofectamine 2000 (Thermo Fisher Scientific, 11668019) were separately diluted in 5 µl Opti-MEM I Reduced Serum Medium (Thermo Fisher Scientific, 31985062). They were then combined into 10 µl and incubated at room temperature for 5 min before being pipetted onto the supernatant in each well. In EYFP reporter experiments, 60 ng EBFP plasmid, 60 ng EYFP* plasmid, 60 ng sgRNA plasmid, and 60 ng ABE8e or 60 ng of each sABE plasmid were transfected. Cells were collected for FACS 48 h after rapamycin addition. In the repurposed EYFP reporter R-loop experiment, 60 ng EBFP plasmid, 60 ng EYFP* plasmid, 60 ng dSaCas9 plasmid, and 60 ng SaCas9 sgRNA plasmid were transfected. In genomic editing experiments, 75 ng sgRNA plasmid and 225 ng ABE8e or 225 ng of each sABE plasmid were transfected. In the R-loop assay, 100 ng SpCas9 sgRNA plasmid, 150 ng ABE8e or 150 ng of each sABE plasmid, 100 ng SaCas9 sgRNA plasmid, and 200 ng dSaCas9 plasmid were transfected. 1 µl 10 µM rapamycin was added to each well in the induction group 10-16 h after transfection. Genomic DNA was extracted 72 h after induction.

### Genomic DNA extraction

The media of each well was gently aspirated. Next, 100 µl freshly prepared lysis buffer [10 mM Tris-HCl, pH 7.5, 0.05% SDS, 25 µg/ml proteinase K (Thermo Fisher Scientific, EO0491)] was added to each well and was incubated at 37 °C for 15 min. The cell lysate was then heat-inactivated at 80 °C for 30 min and used immediately or stored at 4 °C.

### Fluorescence-activated cell sorting

Fourty-eight hours post induction in the EYFP reporter experiments, the media of each well was gently aspirated. 100 µl TrypLE Express (Thermo Fisher Scientific, 12608-028) was added to each well and was incubated at room temperature for 5 min to detach cells. 200 µl complete media was then added to each well, and the mixture was pipetted 30 ~ 40 times for cell suspension. Flow cytometry was carried out on SA3800 Spectral Cell Analyzer (Sony Biotechnology), and the data was analyzed using FlowJo 10.8.1 (FlowJo, LLC). Live cells were gated by side scatter area versus forward scatter area (SSC-A vs. FSC-A). Singlets were selected by forward scatter height versus forward scatter area (FSC-H vs. FSC-A). The fluorescence-Positive population was gated against the mock-transfected control.

### Targeted amplicon sequencing and data analysis

The genomic region flanking each targeted locus was amplified, purified, quantified, and sent for Sanger sequencing (Epoch Life Science) or high-throughput sequencing (HTS) (Amplicon-EZ, Genewiz). Amplicon and primer sequences are available in the shared Benchling links in Supplementary Data. 2, 3, and 4. Partial Illumina adapters provided by Amplicon-EZ were added to the 5′ end of each forward and reverse primer. A unique 6–8 bp barcode was added between the Illumina adapter and the genome-binding sequence to distinguish amplicons from different repeats or conditions pooled in the same sample. PCR was performed in a 10 µl system with 50 pmol of each primer, 10 ~ 50 ng template, and 5 µl 2 × Phanta Max master mix (Dye Plus, Vazyme). The annealing temperature was set to 60 °C, and 35 cycles were used for amplification. The desired DNA fragment was primarily extracted using buffer PB (Qiagen, 19066) when a single and clear band was expected or extracted using QIAquick PCR & Gel Cleanup Kit (Qiagen, 28506) after gel electrophoresis. For Sanger sequencing, each amplicon was eluted in 10 µl ultrapure water (Millipore) and quantified by NanoDrop One (Thermo Fisher Scientific). The Sanger sequencing premix was prepared by adding 1.5 µl eluted DNA (~ 40 ng) and 2.5 µl 10 µM sequencing primer into 9 µl ultrapure water.

For amplicon HTS, multiple different amplicons were pooled together, then purified and eluted in 50 µl ultrapure water. 25 µl eluted DNA was sent in for Amplicon-EZ. Sanger sequencing results were analyzed using EditR (version 1.0.0, https://github.com/MoriarityLab/EditR). Amplicon HTS results were analyzed using CRISPResso2 (version 2.2.9, https://github.com/pinellolab/CRISPResso2).

### RNA-seq experiment

Low-passage HEK293T cells were seeded at $5 \times 10^6$ cells per 10 ml complete media per 10-cm cell culture dish 16 h before transfection. 12 µg construct plasmid was added to 260 µl serum-free DMEM in a 50-tube, followed by the addition of 78 µl PEI Max (1 mg/ml pH = 7.1, Polysciences, 24765-100). The mixture was vortexed and incubated at room temperature for 10 min and then was diluted into a 10 ml complete media. This media replaced the old one in the 10 cm culture dish. 12 h after transfection, 1 µl 1 mM rapamycin was added to each 10-cm plate in the induction group. 48 h after transfection when cells were transfected with ABE8e-P2A-EGFP or nCas9-P2A-EGFP plasmid, or 48 h after induction when cells were transfected with the All-In-One sABE v3.22 plasmid, cells from each 10-cm plate were dissociated with 2 ml of TrypLE Express, centrifuged at $400 \times g$ for 3 min at room temperature, and resuspended in 5 ml complete media. 0.5 to $0.7 \times 10^6$ cells with the top 5% GFP signal were sorted using the MA900 multi-application cell-sorter (Sony). Live cells were gated by the back scatter area versus the forward scatter area (BSC-A vs. FSC-A). Singlets were selected by forward scatter height versus forward scatter area (FSC-H vs. FSC-A). The fluorescence-positive population was gated against the mock-transfected control. RNA was extracted from the sorted cells using the E.Z.N.A Total RNA Kit (Omega Bio-Tek, M6399-00). A quarter of the sorted cells were collected in a separate tube for genomic DNA extraction and on-target DNA base editing analysis. RNA samples were submitted to the Cancer Genomics Center at the University of Texas Health Science Center at Houston (CPRIT RP180734). Total RNA was quality-checked using Agilent RNA 6000 Pico kit (#5067-1513) by Agilent Bioanalyzer 2100 (Agilent Technologies, Santa Clara, USA). RNA with an integrity number >7 was used for library preparation. Libraries were prepared with NEBNext Poly(A) mRNA Magnetic Isolation Module (E7490L, New England Biolabs), NEBNext Ultra II Directional RNA Library Prep Kit for Illumina (E7760L, New England Biolabs), and NEBNext Multiplex Oligos for Illumina (E6609S, New England Biolabs) following the manufacturer's instructions. The qualities of the final libraries were examined using Agilent High Sensitive DNA Kit (#5067-4626) by Agilent Bioanalyzer 2100 (Agilent Technologies, Santa Clara, USA), and the library concentrations were determined by qPCR using Collibri Library Quantification kit (#A38524500, Thermo Fisher Scientific). The libraries were pooled evenly and went for the paired-end 75-cycle sequencing on an Illumina NextSeq 550 System (Illumina, Inc., USA) using High Output Kit v2.5 (#20024907, Illumina, Inc., USA).

### RNA-seq data analysis

RNA-Seq data analysis was conducted according to the GATK Best Practices for RNA-seq short variant discovery (https://github.com/broadinstitute/gatk). Briefly, the RNA-Seq reads were first mapped to GRCh38 using STAR aligner (version 2.7.10a https://github.com/alexdobin/STAR) in two-pass mode with default parameters. Next, the Picard tool MarkDuplicates (version 2.27.4) was applied to mark duplicates in the sorted and mapped BAM files. The refined BAM files were subject to the GATK SplitNCigarReads tool (version 4.2.6.1), which splits reads that contain Ns in their cigar string because they span splicing junctions. Next, GATK BaseRecalibrator (version 4.2.6.1) was used to generate a recalibration table for Base Quality Score Recalibration (BQSR). Known variants in dbSNP version 151 were used during BQSR. Finally, BQSR was applied, and "Analysis-Ready" BAM files were generated. Variant calling was done by GATK HaplotypeCaller (version

4.2.6.1) using default settings with an additional setting to not use the soft-clipped base. SNP variants were filtered out using the GATK selectVariant (version 4.2.6.1). Filters recommended by the GATK for variant calling on RNA-Seq data were used to hard-filtrate qualified variants. Clusters of more than three SNVs identified within a 35-bp window were filtered to maintain high-confidence variants. Hard fitering was applied to select qualified variants with QualByDepth >2.0, FisherStrand <30.0, StrandOddsRatio <3.0, RMSMappingQuality >40.0, MQRankSum >-12.5, ReadPosRankSum >-8.0, and QUAL > 30. The downstream analyses focused only on variants on canonical (1 - 22, X, Y, and M) chromosomes. A-to-G variants were selected, and bam-readcount (version 1.0.1 https://github.com/genome/bam-readcount) was used to quantify the per-base nucleotide abundances per A-to-G variant.

### Inducible knockout experiments

Transfection was performed according to the dosage and method for genomic editing experiments (Described in the transfection method section). 72 h after the media change, the media was gently aspirated. Cells were detached with 100 μl TrypLE Express (Thermo Fisher Scientific) and were incubated at room temperature for 5 min. Next, 500 μl complete media was added to each well. The cell suspension was pipetted firmly 5 - 10 times before being transported to and cultured in 24-well treated tissue culture plates (Genesee Scientific, 25-107). Four days after the transfer, the media was gently aspirated. Cells were detached with 500ul TrypLE Express and were incubated at room temperature for 5 min. The suspended cells were pipetted firmly 5 - 10 times, transferred to 1.5 ml microcentrifuge tubes, and centrifuged at $500 \times g$ for 3 min. The supernatant was discarded, and 500 μl cell staining buffer (BioLegend, 420201) was used to resuspend the cells. Cells were counted by Countess II FL (Thermo Fisher Scientific), and $2-3 \times 10^5$ cells were diluted in 100 μl cell staining buffer. 3 μl of 200 μg/ml FITC anti-human CD46 antibody (BioLegend, Catalog 315304, Lot B339203, Clone MEM-258) or 3 μl of 150 μg/ml PE/Cy7 anti-human β2-microglobulin antibody (BioLegend, Catalog 316318, Lot B371988, Clone 2M2) was mixed with the 100 μl cell suspension and was incubated in the dark on ice for 20 min. The supernatant was discarded, and the cells were washed with 500 μl cell staining buffer by centrifugation at $500 \times g$ for 3 min. The final cell pellet was suspended in 500 μl cell staining buffer. FACS was performed using the SA3800 Spectral Cell Analyzer (Sony Biotechnology). Data were analyzed as described in the fluorescence-activated cell sorting section.

### AAV production and transduction for dual-AAV in vitro editing

Low-passage HEK293T cells were seeded at $5 \times 10^6$ cells per 10 ml complete media per 10-cm cell culture dish (Greiner Bio-One) 16 h before transfection. 3 μg of transfer vector, 5 μg of pHelper plasmid (Cell Biolabs), and 4 μg of AAV-Rep-Cap plasmid (Addgene #112862) were added to 260 μl of serum-free DMEM in a 50-ml tube, followed by addition of 78 μl PEI Max (1 mg/ml PH = 7.1, Polysciences). The mixture was vortexed and incubated at room temperature for 10 min and then was diluted in a 10 ml complete media. This media replaced the old one in the 10-cm culture dish. 48 h after transfection, all supernatant was collected in a 15-ml tube and centrifuged at $3200 \times g$ for 5 min at room temperature to remove the cell debris. The supernatant was then concentrated using a PEG virus precipitation kit (Biovision, K904-50) with an optimized protocol. Briefly, 2.5 ml PEG solution was added to the supernatant. The mixture was inverted evenly, refrigerated at 4 °C for 24 h, and then centrifuged at $3200 \times g$ and 4 °C for 30 min. Several aspiration and centrifugation rounds were applied to remove the supernatant from the pellet entirely. The freshly prepared AAVs were used immediately for transduction. For AAV transduction, HEK293T cells were seeded at 1,500 cells per 100 μl complete media per well in the 96-well Poly-D-lysine plate (Corning, 356690) and the cells were incubated at room temperature for 15 min. Then, 10 μl of

each AAV vector was added to each well, and transduced cells were cultured at 37 °C with 5% CO2. Genomic DNA extractions were performed on day 7 after transduction.

### Lentivirus production and transduction for m*PCSK9* HEK293T model

A 200-bp DNA fragment containing the m*PCSK9* genomic locus of interest was amplified from the C57BL/6 mouse genome and was ligated to the lentiviral transfer plasmid (Addgene #52963) through Golden Gate assembly. Low-passage HEK293T cells were seeded at $2 \times 10^4$ cells per 100 μl complete media per well on a 96-well Poly-D-lysine plate (Corning) 16 h before transfection. They were incubated at room temperature for 15 min before transferring into the 37 °C, 5% CO2 incubator. For each well, 111 ng of transfer plasmid, 83 ng of packing plasmid psPAX2 (Addgene, #12260), and 56 ng of envelope plasmid pMD2.G (Addgene, #12259) were added to 5 μl Opti-MEM I Reduced Serum Medium (Thermo Fisher Scientific). The mixture was combined with another 5 μl Opti-MEM solution containing 0.5 μl Lipofectamine 2000 (Thermo Fisher Scientific) and incubated at room temperature for 5 min before being added to the well. 48 h after transfection, the supernatant was collected and centrifuged at $3000 \times g$ for 5 min at room temperature. Low passage cells were seeded at 100,000 cells per 500 μl complete media per well in a 24-well plate (Genesee Scientific, 25-107) and incubated at room temperature for 15 min. 5 μl lentivirus-containing supernatant was added to the well on the 24-well plate, and the cells were cultured in 37 °C, 5% CO2 incubator. 24 h after transduction, the old culture media was replaced by 500 μl fresh complete media with 1 μg/ml puromycin (Gibco, A1113802) to initiate puromycin selection. When the surviving cells reached 80% confluency, they were dissociated with 200 μl/well TrypLE Express (Thermo Fisher Scientific) and added to 10 ml complete media containing 1 μg/ml puromycin in a 10-cm plate (Greiner Bio-One) for further proliferation. The stable cell line was verified by targeted genomic DNA PCR amplification followed by Sanger sequencing. The verified stable cell line was cryo-stored until use.

### AAV and lentivirus production and transduction for triple-AAV or triple-lentiviral vector in vitro editing

$2 \times 10^6$ HEK293T cells were seeded into 10-cm dishes (Greiner Bio-One) in 15 ml of complete media. When cells reached 30% confluency, for AAV production, 3 μg of pAAV2/2 plasmids (Addgene #104963) and 3 μg of pAdDeltaF6 plasmids (Addgene #112867) were mixed with 3 μg of transgene plasmids, 4 ml DMEM, and 60 μl PEI (1 mg/ml pH = 7.1, Polysciences). For lentivirus production, 3 μg of PspAX2 (Addgene #12260), 3 μg of pMD2.G (Addgene #12259), and 3 μg of transgene plasmids were mixed with 4 ml DMEM and 60 μL PEI (1 mg/ml PH = 7.1, Polysciences). The mixture was incubated for 20 min at room temperature before being added to the cell culture. Twenty-four hours after transfection, the old media was replaced by 15 ml fresh complete media. 72 h (or 48 h for lentivirus production) after media change, the cell culture medium was transferred to a 50 ml conical tube. Cells were dissociated with trypsin-EDTA (0.25%) (Gibco, 25200056) and transferred to the same tube. DMEM was added to achieve a final volume of 30 ml. 3 ml chloroform was added, and the mixture was vortexed for 5 min. Next, 7.6 ml of 5 M NaCl was added, and the mixture was vortexed for 10 s before centrifugation at $3000 \times g$ for 5 min at 4 °C. The aqueous phase was collected, and 9.4 ml 50% (v/v) PEG 8000 (Fisher BioReagents, BP2331) was added. This mixture was vortexed for 10 s and incubated at 4 °C overnight. The next day, it was centrifuged at $3000 \times g$ for 30 min at 4 °C. The cell pellet was resuspended with 700 μl PBS buffer (Gibco, 10010-023). 1 μl Cryonase Cold-active nuclease (TaKaRa #2670 A) and 1.75 μl 1 M MgCl₂ were added to each tube, and the mixture was incubated at room temperature for 30 mins. 700 μl chloroform was added, then the mixture was vortexed for 10 s and centrifuged at $3000 \times g$ for

5 min at 4 °C. The virus-containing aqueous phase was either used immediately or stored at 4 °C.

HEK293T cells were seeded in 96-well plates (Corning) at 10% confluency for transduction. 12 h post seeding, 10 μL of each AAV virus or lentivirus was mixed and added into the culture medium. For lentivirus transduction, 5 μg/mL polybrene (Sigma #TR1003G) was supplied into the cell media. The media was replaced with fresh complete media after 24 h. Three days later, cells were detached and transferred to a 24-well plate (Genesee Scientific). After another 4 days, genomic DNA was extracted.

## AAV production for animal studies

High-purity AAV viruses with AAV2 inverted terminal repeat pseudotyped with AAV8 capsid were produced by the Gene Vector Core at the Baylor College of Medicine. The titers of AAV viruses were measured by real-time qPCR.

## Animal studies

A total of twelve C57BL/6 male mice used in animal experiments were purchased from the Jackson Laboratory. They were maintained and handled following laboratory animal treatments approved by the Institutional Animal Care and Use Committee (IACUC) of Baylor College of Medicine (BCM). All mice were housed in an animal facility with standard conditions such as pathogen-free, light-dark cycle (12 h:12 h), 22--25 °C air temperature, and 40-70% air humidity on 2920X Teklad Global Extruded Rodent Diet (Soy Protein-Free; Harlan Laboratories). At 15 weeks of age, mice were randomly grouped into three groups of four each and subjected to the experimental treatments. Specifically, mice in the experimental groups received $1 \times 10^{11}$ genome copy (gc) per AAV vector buffered in 200 μl sterile saline via tail-vein injection. Three days after the AAV injection, mice in the induction group were injected with 3 mg/kg rapamycin buffered in the vehicle [a mixture of equal volume 10% PEG400 (MiliporeSigma, 8.07485.1000) and 10% Tween 80 (MiliporeSigma, 1754-25 ML)] every other day for 8 days through intraperitoneal injection (i.p.). Mice in the other groups were injected with the same volume of vehicle. Six days after the final injection of rapamycin, mice were euthanized, and 50 mg of mouse liver tissue was homogenized in 600 μl DPBS (Corning, 21-031-CV). The homogenate was then centrifuged at 2000 × g for 5 min at 4 °C, and the pellet was lysed using 600 μl lysis buffer [10 mM Tris-HCl, pH 7.5, 0.05% SDS, 25 μg/ml proteinase K (Thermo Fisher Scientific)] and incubated at 65 °C for 15 min, 68 °C for 15 min, and 98 °C for 10 min.

## Statistics and reproducibility

All bar plots were created with dots indicating individual biological replicates. When there were more than two replicates, error bars were used to represent standard deviation. Groups were compared using either multiple unpaired two-tailed $t$-tests or unpaired two-tailed $t$-tests, with significance notations as ns (not significant), $*P < 0.05$, $**P < 0.01$, $***P < 0.001$, and $****P < 0.0001$. Relevant statistical details can be found in figure legends or descriptions. No statistical method was used to predetermine the sample size. Sample sizes were determined by observed variability across independent experiments, and no data were excluded from the analyses. These sizes align with standard practices in related research. For animal experiments, at 15 weeks of age, the twelve C57BL/6 male mice were randomly divided into three groups, with four mice in each group. The investigators were not blinded to allocation during experiments and outcome assessment, as the data were processed and analyzed in exactly the same way, and there were no subjective decisions or interpretations made during the data analysis phase.

## Reporting summary

Further information on research design is available in the Nature Portfolio Reporting Summary linked to this article.

## Data availability

High-throughput DNA- and RNA-Seq data generated in this study have been deposited at the Sequence Read Archive PRJNA923001. Data presented in each figure are provided in Source Data. Nucleic acid sequences of all constructs are provided in the the Supplementary Data. 1. Nucleic acid sequence of genomic loci tested and primers used in this study are provided in Supplementary Data. 2, 3, and 4. The structure of TadA-8e can be found in Protein Data Bank PDB: 6VPC[38] [https://www.rcsb.org/structure/6vpc]. Source data are provided with this paper.

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

## Acknowledgements

This work was supported by the National Science Foundation CAREER AWARD (CBET-2143626 to X.G.); Robert A. Welch Foundation grant (C-1952 to X.G.); National Institutes of Health grants (HL157714 to X.G.; DK111436 and AG069966 to Z.S.); the DLDCCC (P30CA125123 to Z.S.), the Specialized Programs of Research Excellence (SPORE) program (P50CA126752 to Z.S.), the Gulf Coast Center for Precision Environmental Health (P30ES030285 to Z.S.), and the Texas Medical Center Digestive Diseases Center (P30DK056338 to Z.S.). We thank the Cancer Genomics Center at The University of Texas Health Science Center at Houston for performing the RNA-sequencing service. We thank Harshavardhan Deshmukh at Rice University Shared Equipment Authority for support in FACS analysis. Figure 1d is created with PyMOL[71]. Figure 5f is created with BioRender.com.

## Author contributions

H.Z., Q.Y., D.M., F.P., Z.S., and X.G. designed the research. H.Z., Q.Y., D.M., F.P., A.L., K.C., P.G., and E. C. O. performed the experiments. H.Z., Q.Y., and D.M. analyzed the data. H.Z. performed computational analysis of RNA-Seq data. H.Z. wrote the initial draft. H.Z., Q.Y., Z.S., and X.G. revised the manuscript with help from all authors. X.G. and Z.S. supervised the project.

## Competing interests

X.G. and H.Z. are in the process of filing a provisional patent application on this work. The remaining authors declare no competing interests.
