## [Peer Review File · Nature Communications]

Reviewers' Comments:

Reviewer #1:

Remarks to the Author:

SUMMARY. Zeng, Yuan, Peng & Ma et al. have developed and characterized an inducible, split-deaminase ABE8 platform (sABE v3.22) to increase genomic and transcriptomic specificity of adenine base editing while preserving on-target editing efficiency. They engineered different variants of a split TadA-8e and use chemically induced dimerization (CID) for rapamycin-based control over deaminase assembly and function. Interestingly, their sABE also has a much narrower editing window and, importantly, a much higher allelic outcome of single base substitutions than WT-ABE8e. Moreover, the authors show that both gRNA-dependent and independent genomic off-target editing and gRNA-independent RNA off-target editing are substantially reduced with sABE. Finally, the authors show how this system can be used for inducible multiplex knock-outs via splice donor disruption in human cells and they deliver sABE in a dual-AAV setup to the mouse liver to demonstrate efficient CID-based adenine base editing *in vivo*.

ASSESSMENT. The authors report findings that are highly relevant to the gene editing field and beyond. The study is well-executed and the data look robust. The figures are also well made and easy to understand. A thoroughly engineered CID-based ABE8 platform is important, given ABE8 is the clinically most important BE system to date. In the light of recently reported findings from Yan et al. in *Nature Communications* (PMID 36997536), it seems favorable, not to deliver an intact deaminase (especially a highly active one like TadA-8e) via AAV for relatively long expression in the recipient cell/tissue. All aspects that I've mentioned in the summary (narrowed editing window, "true" single base editing, reduced off-target editing, preserved on-target functionality, preclinical applications) make this work a valuable contribution for the gene editing field and a broader audience, given the widespread use of these technologies in the biomedical community. The robust *in vivo* efficiency of the system is very encouraging, too, especially given the relatively low total dose of 2×10^{11} AAV vector genomes per mouse.

In sum, this is exciting work! Congrats to the authors on this manuscript.

MAJOR

1. It would be great, if the authors could adapt their split-TadA_8e approach to a TadA-based CBE architecture, such as TadCBE. A quick characterization on a couple of sites in HEK293T would be helpful. If this system can be expanded to CBEs, it would further increase its impact.
2. Given the hypothesized importance of reduced/shortened cellular exposure to deaminase expression, it would be great so see if deaminase activity/assembly can be turned off again by removing rapamycin? Maybe one could test this in HEK293T cells using either the EYFP reporter or by tracking some (reversible) high-frequency RNA off-target edits (no sorting or RNA-seq needed). This would allow to observe deaminase activity over time AFTER removal of rapamycin?

MINOR

3. One might consider citing Yan et al. regarding potential off-target effects arising due to long-term expression of BEs *in vivo* (PMID 36997536).

Reviewer #2:

Remarks to the Author:

Zeng et al. described a split adenine base editor (sABE) that utilizes chemically induced dimerization (CID) to control the catalytic activity of the deoxyadenosine deaminase TadA-8e. This technique shows high on-target editing activity and low background activity, compared to the original ABE with TadA-8e (ABE8e). The authors further demonstrate that the optimized version sABE v3.22, achieves high DNA on-target editing efficiencies and multiplex gene knockout in

HEK293T cells. Finally, they also performed in vivo editing of the mouse gene PCSK9 by delivering dual AAVs encoding sABE v3.22.

However, the potential application of sABE v3.22 is unclear, as the manuscript only evaluates the efficiency of A-to-G conversion in genomic DNA from mouse live tissue. Furthermore, other biomarkers in this model, such as the concentration and function of the PCSK9 protein and the serum LDL level, are not shown. Here are some suggestions for the authors to further improve the quality of the work.

Major concerns:

1. The authors claimed that their sABE v3.22 system has "high precision and low off-target effect" compared to other ABEs. However, to further support this claim, the authors should compare sABE v3.22 with other ABEs that belong to "post-translational inducible" rather than the non-inducible ABE8e.
2. To further strengthen the study, it would be valuable for the authors to include a comparison between sABE v3.22 and the miniABEs discussed in the paper "TadA reprogramming to generate potent miniature base editors with high precision." Alternatively, the authors could discuss the advantages and disadvantages of sABE compared to miniABE.
3. The detection of only two genes in result 4 seems inadequate. The authors should consider testing more genes to strengthen their results.
4. The authors asserted that their sABE system could address the risks related to prolonged expression of active ABE8e, but they did not specify these risks. It would be valuable for the authors to provide a list of these specific risks and demonstrate, through experimental evidence, how their system mitigates these risks. These experimental data are important to support the usefulness of their ABA-inducible adenine base editor tool.
5. The authors should study the physiological function of the mice after editing of the PCSK9 gene by the inducible base editor.
6. The authors should provide more detailed experimental methods, such as the flow cytometry protocol, to improve the clarity of their study.

Minor concerns:

1. In line 86, it is stated that "We constructed sABE v1 and v2...loop-25 or loop-74 of the TadA-8e deaminase domain, respectively." However, the use of "or" and "respectively" together in this sentence is confusing. The authors should rephrase the sentence to clarify the intended meaning.
2. In Supplementary Figure 1, the Y-axis label in the middle column should be "FSC-H" instead of "SSC-A"?
3. In Supplementary Figure 2, the authors should explain what "BSC-A" represents in the Y-axis label for clarity, or provide more detailed experimental methods about this Figure.
4. Please thoroughly check all spellings, images, etc. in the document.

Reviewer #3:

Remarks to the Author:

The development of base editors (BEs) is a rapidly advancing area of research that is being hotly pursued for use in targeted gene therapy. The simplest BEs consist of a nicking Cas protein connected to a deaminase by a flexible linker. Target-site specificity is achieved by coexpression of an appropriate sgRNA. However, the therapeutic use of BEs is confounded by some undesirable properties: a) modification of "bystander" bases adjacent to the targeted site, b) modification of Cas/sgRNA off-target sites, c) modification of bases at fortuitously accessible loci throughout the genome and d) modification of bases throughout the transcriptome. In this paper, Zheng et al. describe the development and testing of a split ABE (adenine base editor) that can be induced to dimerise and thus become active upon binding rapamycin. Their "final product", sABE8e v.22, represents a huge improvement over non-regulatable ABEs. It has low activity in the absence of rapamycin, reduced modification of bystander bases, fewer off-target modifications and reduced modification of the transcriptome.

The approach is similar to that described by Berrios et al. 2021 for cytosine base editors, but since

Zheng et al. used a different deaminase they were practically starting from scratch. In my estimation, this paper would have as much or more impact than a recent Nat Comm paper on ABE8e: doi.org/10.1038/s41467-022-35508-7

The data are robust, internally consistent and presented in standard format for the field. Statistical analyses are provided, but the overall conclusions are evident simply by visual inspection of the data.

Standard methods are used. Usage of an EYFP reporter for monitoring editing efficiency is a nice touch.

The current manuscript already encompasses a large body of work. If I had to request another experiment, I would ask whether incorporating F148A would further reduce off-target transcriptome editing.

The manuscript is generally clear and easy to understand if you don't mind looking up some of the cited literature.

The authors appear to have considered the published literature fairly effectively. I think it would be useful for them to discuss why one would use sABE v3.22 instead of split prime editor, which has no detectable off-targets. Cite and compare with doi.org/10.1038/s41392-022-01234-1

Also for discussion: what level of knockdown of targets B2M, CD46 and PCSK8 is necessary to achieve therapeutic effects?

Some minor points:

In the title ABE is referred to as an adenine base editor, but throughout most of the manuscript it is termed an adenosine editor. Be consistent.

Figure 1.

Diagram of R loop region would be more clear if nick were shown in strand that is bound by nCas9 and if deaminated base were aligned with adjacent bases. As it is, it appears that the edited strand is nicked.

Figure 3. The R-loop assay diagram is not easy to understand unless you are familiar with Doman et al. 2020. I suggest that the authors include a brief explanation of the technique, similar to that provided by Berrios et al., 2021. The terminology in the text and figure needs to be treated carefully when discussing the R-loop assay. The authors should clarify that while the assay is used to provide a measure of Cas9/sgRNA independent off-target effects, the method itself is dependent on both Cas9 (dSaCas9) and sgRNA (off-target).

Fig 5 - in legend define IntN and IntC.

Extended data Figure 7. In legend define abbreviations bGHpA and WPRE.

Supplementary file Benchling links didn't work for me. Are these just placeholders?

Point-to-point response to reviewer comments

Reviewer #1 (Remarks to the Author):

SUMMARY. Zeng, Yuan, Peng & Ma et al. have developed and characterized an inducible, split-deaminase ABE8 platform (sABE v3.22) to increase genomic and transcriptomic specificity of adenine base editing while preserving on-target editing efficiency. They engineered different variants of a split TadA-8e and use chemically induced dimerization (CID) for rapamycin-based control over deaminase assembly and function. Interestingly, their sABE also has a much narrower editing window and, importantly, a much higher allelic outcome of single base substitutions than WT-ABE8e. Moreover, the authors show that both gRNA-dependent and independent genomic off-target editing and gRNA-independent RNA off-target editing are substantially reduced with sABE. Finally, the authors show how this system can be used for inducible multiplex knock-outs via splice donor disruption in human cells and they deliver sABE in a dual-AAV setup to the mouse liver to demonstrate efficient CID-based adenine base editing *in vivo*.

ASSESSMENT. The authors report findings that are highly relevant to the gene editing field and beyond. The study is well-executed and the data look robust. The figures are also well made and easy to understand. A thoroughly engineered CID-based ABE8 platform is important, given ABE8 is the clinically most important BE system to date. In the light of recently reported findings from Yan et al. in *Nature Communications* (PMID 36997536), it seems favorable, not to deliver an intact deaminase (especially a highly active one like TadA-8e) via AAV for relatively long expression in the recipient cell/tissue. All aspects that I've mentioned in the summary (narrowed editing window, "true" single base editing, reduced off-target editing, preserved on-target functionality, preclinical applications) make this work a valuable contribution for the gene editing field and a broader audience, given the widespread use of these technologies in the biomedical community. The robust *in vivo* efficiency of the system is very encouraging, too, especially given the relatively low total dose of 2×10^{11} AAV vector genomes per mouse. In sum, this is exciting work! Congrats to the authors on this manuscript.

A few comments should be addressed:

1. It would be great, if the authors could adapt their split-TadA_8e approach to a TadA-based CBE architecture, such as TadCBE_d. A quick characterization on a couple of sites in HEK293T would be helpful. If this system can be expanded to CBEs, it would further increase its impact.

Response: We appreciate the kind compliments and valuable suggestions. We have been thinking precisely the same thing - to adapt our approach to TadCBE_d. We have recently obtained some encouraging preliminary data on split TadCBE_d. We plan to present the results in a separate manuscript since we also want to test how they would work *in vivo*. In addition, we constructed and compared the performance of sABE variants bearing beneficial mutations, including V106W and F148A. We found that the sABE v3.22 architecture is compatible with these variants. We have added these new data to the revised manuscript.

2. Given the hypothesized importance of reduced/shortened cellular exposure to deaminase expression, it would be great so see if deaminase activity/assembly can be turned off again by removing rapamycin? Maybe one could test this in HEK293T cells using either the EYFP reporter or by tracking some (reversible) high-frequency RNA off-target edits (no sorting or

RNA-seq needed). This would allow to observe deaminase activity over time AFTER removal of rapamycin?

Response: We have performed experiments to explore the effect of rapamycin removal. We found that the sABE v3.22 activity can be blunted after rapamycin removal. These new data have been added to the revised manuscript. Thank you for the suggestion.

3. One might consider citing Yan et al. regarding potential off-target effects arising due to long-term expression of BEs in vivo (PMID 36997536).

Response: We have included this reference and a brief description in the revised manuscript.

Reviewer #2 (Remarks to the Author):

Zeng et al. described a split adenine base editor (sABE) that utilizes chemically induced dimerization (CID) to control the catalytic activity of the deoxyadenosine deaminase TadA-8e. This technique shows high on-target editing activity and low background activity, compared to the original ABE with TadA-8e (ABE8e). The authors further demonstrate that the optimized version sABE v3.22, achieves high DNA on-target editing efficiencies and multiplex gene knockout in HEK293T cells. Finally, they also performed in vivo editing of the mouse gene PCSK9 by delivering dual AAVs encoding sABE v3.22.

However, the potential application of sABE v3.22 is unclear, as the manuscript only evaluates the efficiency of A-to-G conversion in genomic DNA from mouse live tissue. Furthermore, other biomarkers in this model, such as the concentration and function of the PCSK9 protein and the serum LDL level, are not shown. Here are some suggestions for the authors to further improve the quality of the work.

A few comments should be addressed:

1. The authors claimed that their sABE v3.22 system has "high precision and low off-target effect" compared to other ABEs. However, to further support this claim, the authors should compare sABE v3.22 with other ABEs that belong to "post-translational inducible" rather than the non-inducible ABE8e.

Response: We appreciate the comments and suggestions. Our sABE is the first "post-translationally inducible" ABE, to our knowledge. Therefore, comparing our inducible sABE v3.22 with the non-inducible ABE8e is essential. This comparison highlights the potential advantages of incorporating post-translational regulation into ABE systems, as evidenced by the enhanced precision and reduced off-target effects we observed. We have revised the manuscript to clarify that "sABE v3.22 has higher precision and lower off-target effect than the unregulated ABE8e".

2. To further strengthen the study, it would be valuable for the authors to include a comparison between sABE v3.22 and the miniABEs discussed in the paper "TadA reprogramming to generate potent miniature base editors with high precision." Alternatively, the authors could discuss the advantages and disadvantages of sABE compared to miniABE.

Response: We appreciate this suggestion. Our sABE v3.22 specifically targets the regulation of TadA, as opposed to the Cas effector in miniABEs. We have constructed and tested split TadA

systems with SaCas9 nickase and engineered dead Un1Cas12f1 variants, namely CasMINlv3.1, the same Cas effector used in miniABEs, and CasMINlv4. We found that sABE v3.22 architecture is compatible with other Cas effectors. These new data have been included in the revised manuscript. We have also cited related research articles, including the one you mentioned.

3. The detection of only two genes in result 4 seems inadequate. The authors should consider testing more genes to strengthen their results.

Response: We understand the concern. We are constantly optimizing the system. The two genes served as a proof-of-concept for multiplex inducible gene knockout using ABE. We selected Beta-2 microglobulin (B2M) and CD46 regulatory proteins because (1) they are important targets for allogeneic cell therapies and cancer research; (2) we have worked out the multiplex base-editing strategy in our previous work (PMID: 35589728). We think that the data from these two genes clearly show that sABE can be successfully applied to edit multiple splicing sites of a gene simultaneously and in a controlled manner. We are working to make using the system for other genes more convenient. We have added these clarifications in the revised manuscript.

4. The authors asserted that their sABE system could address the risks related to prolonged expression of active ABE8e, but they did not specify these risks. It would be valuable for the authors to provide a list of these specific risks and demonstrate, through experimental evidence, how their system mitigates these risks. These experimental data are important to support the usefulness of their ABA-inducible adenine base editor tool.

Response: We appreciate the opportunity to clarify this important question. Our system uses rapamycin-induced FKBP-FRB CID to regulate the activity of TadA-8e deaminase rather than using the ABA-induced ABI-PYL CID system. We have added the risks related to the prolonged expression of active base editors, including CBEs and ABEs, which were recently reported by Yan et al. in Nature Communications (PMID: 36997536). As noted from the literature, such analyses would take a long time (15 months) and multiple whole-genome sequencing. Although we did not directly compare ABE8e and sABE v3.22 for a prolonged period, we have added new data showing that the sABE v3.22 activity can be blunted after rapamycin removal. Therefore, given the inducible nature of our sABE system and the improved editing outcomes compared to ABE8e in the short run, it is conceivable that our inducible ABE tool would mitigate the risks associated with the prolonged expression of active base editors. We have tuned down the related statements in the revised manuscript.

5. The authors should study the physiological function of the mice after editing of the PCSK9 gene by the inducible base editor.

Response: We appreciate this suggestion. Since rapamycin is known to upregulate PCSK9 (PMID: 22426206), the PCSK9 protein-level reduction or cholesterol-lowering effects of the rapamycin-dependent sABE system would be underestimated. The downstream physiological outcome of the efficient PCSK9 gene editing has been widely validated (PMID: 34012094, 35902773, 34012082). Therefore, the current study focuses on using PCSK9 as a proof-of-concept for the sABE v3.22 system with the DNA-level editing as a final readout. We are developing and testing rapamycin-independent sABE systems for more in-depth functional studies. We have discussed these pitfalls in the revised manuscript.

6. The authors should provide more detailed experimental methods, such as the flow cytometry protocol, to improve the clarity of their study.

Response: We have updated the Methods section and the Supplementary Information to include necessary experimental details related to the flow cytometry protocol. Thanks for the suggestion.

7. In line 86, it is stated that "We constructed sABE v1 and v2....loop-25 or loop-74 of the TadA-8e deaminase domain, respectively." However, the use of "or" and "respectively" together in this sentence is confusing. The authors should rephrase the sentence to clarify the intended meaning.

Response: We have revised the sentence.

8. In Supplementary Figure 1, the Y-axis label in the middle column should be "FSC-H" instead of "SSC-A."?

Response: We have corrected it in this Supplementary Figure. Thank you for pointing it out.

9. In Supplementary Figure 2, the authors should explain what "BSC-A" represents in the Y-axis label for clarity, or provide more detailed experimental methods about this Figure.

Response: We have updated the supplementary information to describe the use of FSC-A vs. BSC-A accurately and provided additional information about the flow cytometry protocol employed in our study using the Sony MA900 flow cytometer.

10. Please thoroughly check all spellings, images, etc. in the document.

Response: We have checked throughout the manuscript and made corrections to spelling, image labeling, and other relevant elements to ensure the clarity and accuracy of our presentation. We also checked the supplementary information and figures.

Reviewer #3 (Remarks to the Author):

The development of base editors (BEs) is a rapidly advancing area of research that is being hotly pursued for use in targeted gene therapy. The simplest BEs consist of a nicking Cas protein connected to a deaminase by a flexible linker. Target-site specificity is achieved by coexpression of an appropriate sgRNA. However, the therapeutic use of BEs is confounded by some undesirable properties: a) modification of "bystander" bases adjacent to the targeted site, b) modification of Cas/sgRNA off-target sites, c) modification of bases at fortuitously accessible loci throughout the genome and d) modification of bases throughout the transcriptome. In this paper, Zheng et al. describe the development and testing of a split ABE (adenine base editor) that can be induced to dimerise and thus become active upon binding rapamycin. Their "final product", sABE8e v.22, represents a huge improvement over non-regulatable ABEs. It has low activity in the absence of rapamycin, reduced modification of bystander bases, fewer off-target modifications and reduced modification of the transcriptome.

The approach is similar to that described by Berrios et al. 2021 for cytosine base editors, but since Zheng et al. used a different deaminase they were practically starting from scratch.

In my estimation, this paper would have as much or more impact than a recent Nat Comm paper on ABE8e: doi.org/10.1038/s41467-022-35508-7

The data are robust, internally consistent and presented in standard format for the field. Statistical analyses are provided, but the overall conclusions are evident simply by visual inspection of the data.

Standard methods are used. Usage of an EYFP reporter for monitoring editing efficiency is a nice touch.

The current manuscript already encompasses a large body of work. If I had to request another experiment, I would ask whether incorporating F148A would further reduce off-target transcriptome editing.

The manuscript is generally clear and easy to understand if you don't mind looking up some of the cited literature.

A few comments should be addressed:

1. If I had to request another experiment, I would ask whether incorporating F148A would further reduce off-target transcriptome editing.

Response: We are thankful for the favorable comparison of our study to another study recently published in Nat Comm. We also appreciate the suggestion to incorporate F148A, although the reviewer does not 'have to request another experiment'. We constructed and compared the performance of sABE variants bearing beneficial mutations, including V106W and F148A. We found that the sABE v3.22 architecture is compatible with these variants. We envision that one would see a synergistic effect on decreasing transcriptomic off-target effects by implementing these mutations. However, due to the limited time for this revision, we did not perform additional RNA-seq experiments. We have added the new data and related discussion in the revised manuscript.

2. I think it would be useful for them to discuss why one would use sABE v3.22 instead of split prime editor, which has no detectable off-targets. Cite and compare with doi.org/10.1038/s41392-022-01234-1

Response: Both prime editing and base editing exploit the programmability of CRISPR-Cas to guide fused effectors to specific genomic loci. Prime editors, such as PE2, employ a reverse transcriptase (RT) to transfer information from the modified guide RNA (pegRNA) to the nicked DNA strand. Base editors, on the other hand, utilize a deaminase to modify either purine or pyrimidine bases on the non-nicked strand, triggering downstream DNA repair mechanisms to achieve base conversion. The exact mechanism by which the edited strand supersedes the original strand during prime editing, despite the latter's better base-pairing, remains unclear. In addition, it is not well understood why cellular DNA repair mechanisms would use the edited strand rather than the unedited complementary strand for DNA repair. To circumvent this issue, PE3 introduces an additional nicking sgRNA to the unedited strand, effectively biasing the cellular repair machinery towards the edited strand. However, this comes with the risk of double-strand breaks (DSBs), thus, a higher chance of unintended indels. By comparison, base editors only nick one DNA strand, circumventing the risk of DSB-associated indels. Thus, base editors offer reliable site-specific nucleotide conversion with less need for guide RNA optimization.

The research article doi.org/10.1038/s41392-022-01234-1 utilized PE3 by splitting Cas9 to

circumvent the packaging capacity limitations of adeno-associated viruses (AAVs). We used a similar split intein approach to deliver our sABE v3.22 system *in vivo*, a common strategy for payloads exceeding the 4.7 kb capacity of AAVs, although we focus on different aspects of the system. We have added related discussions in the revised manuscript, and we have cited this article.

3. Also for discussion: what level of knockdown of targets B2M, CD46 and PCSK8 is necessary to achieve therapeutic effects?

Response: B2M and CD46 are important targets for allogeneic cell therapies and cancer research. The current practice of gene knockout for cell therapies usually happens *ex vivo* in patient-derived T-cells, followed by genotype (and phenotype) selection, clonal proliferation, and transfusion (PMID: 37314354). Therefore, the level of gene knockdown is less of a concern. In terms of PCSK9 knockout for therapeutic purposes, Tanja Rothgangl and coworkers provided a nice relationship between the relative PCSK9 mRNA expression and the targeted adenine editing efficiency (PMID: 34012094). They showed that 50-70% A-to-G base editing, at the same target as we used, can significantly reduce cholesterol in macaques. Consistently, another paper showed that 50-80% A-to-G base editing can reduce cholesterol in primates (PMID: 34012082). Future research is needed to refine the relationship among base editing efficiency, mRNA expression, PCSK9 protein production, and physiological changes. We have added these discussions in the revised manuscript.

4. In the title ABE is referred to as an adenine base editor, but throughout most of the manuscript it is termed an adenosine editor. Be consistent.

Response: We changed the terminology used throughout the manuscript to be consistent, and we used the term "adenine base editor" to accurately reflect the technology employed in our study. Thank you for the suggestion.

5. Figure 1. Diagram of R loop region would be more clear if nick were shown in strand that is bound by nCas9 and if deaminated base were aligned with adjacent bases. As it is, it appears that the edited strand is nicked.

Response: We appreciate the suggestion. We have revised the cartoon in Figure 1 accordingly. The target strand (TS), which sgRNA forms RNA-DNA heteroduplex with, is nicked by the Cas9 nickase.

6. Figure 3. The R-loop assay diagram is not easy to understand unless you are familiar with Doman et al. 2020. I suggest that the authors include a brief explanation of the technique, similar to that provided by Berrios et al., 2021. The terminology in the text and figure needs to be treated carefully when discussing the R-loop assay. The authors should clarify that while the assay is used to provide a measure of Cas9/sgRNA independent off-target effects, the method itself is dependent on both Cas9 (dSaCas9) and sgRNA (off-target).

Response: We agree. We have added a brief explanation of the R-loop assay technique in the main text.

7. Fig 5 - in legend define IntN and IntC.

Response: We have added definitions for "IntN" and "IntC" in the figure legend. "IntN" refers to the N-terminal part of the gp41-1 intein, while "IntC" refers to the C-terminal part of the gp41-1 intein.

8. Extended data Figure 7. In legend define abbreviations bGHpA and WPRE.

Response: We have added these definitions. "bGHpA" refers to the bovine growth hormone polyadenylation signal, while "WPRE" refers to the Woodchuck hepatitis virus posttranscriptional regulatory element.

9. Supplementary file Benchling links didn't work for me. Are these just placeholders?

Response: The issue may have been caused by the conversion to PDF on a Mac. We have generated a new PDF file optimized for electronic distribution. We have tested the new PDF file on a Windows system and confirmed its accessibility.

Reviewers' Comments:

Reviewer #1:

Remarks to the Author:

Thanks for addressing all of my comments and adding new data.
Congrats to the whole team!

Reviewer #2:

Remarks to the Author:

The authors have addressed my raised questions.

Reviewer #3:

Remarks to the Author:

Thanks for your responses and revisions. Looks good to me.

Point-to-point response to reviewer comments

Reviewer #1 (Remarks to the Author):

Thanks for addressing all of my comments and adding new data.
Congrats to the whole team!

Response: We greatly appreciate your positive feedback and thoughtful comments.

Reviewer #2 (Remarks to the Author):

The authors have addressed my raised questions.

Response: Thank you for acknowledging our efforts to address your questions.

Reviewer #3 (Remarks to the Author):

Thanks for your responses and revisions. Looks good to me.

Response: We are pleased to hear that our revision met your expectation.